# Generating Freeform Endoskeletal Robots

**Muhan Li, Lingji Kong, Sam Kriegman**
Northwestern University

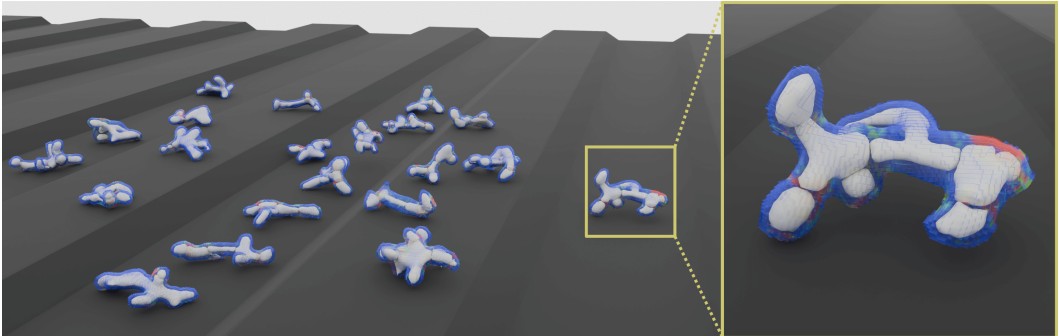

Figure 1: An evolving population of endoskeletal soft robots were encoded by a low dimensional latent design genome with minimal morphological assumptions and optimized for locomotion across complex terrains in multiphysics simulation using a shared universal controller that was simultaneously learned alongside morphological design. Videos and code at https://endoskeletal.github.io.

## Abstract

The automatic design of embodied agents (e.g. robots) has existed for 31 years and is experiencing a renaissance of interest in the literature. To date however, the field has remained narrowly focused on two kinds of anatomically simple robots: (1) fully rigid, jointed bodies; and (2) fully soft, jointless bodies. Here we bridge these two extremes with the open ended creation of terrestrial endoskeletal robots: deformable soft bodies that leverage jointed internal skeletons to move efficiently across land. Simultaneous de novo generation of external and internal structures is achieved by (i) modeling 3D endoskeletal body plans as integrated collections of elastic and rigid cells that directly attach to form soft tissues anchored to compound rigid bodies; (ii) encoding these discrete mechanical subsystems into a continuous yet coherent latent embedding; (iii) optimizing the sensorimotor coordination of each decoded design using model-free reinforcement learning; and (iv) navigating this smooth yet highly non-convex latent manifold using evolutionary strategies. This yields an endless stream of novel species of "higher robots" that, like all higher animals, harness the mechanical advantages of both elastic tissues and skeletal levers for terrestrial travel. It also provides a plug-and-play experimental platform for benchmarking evolutionary design and representation learning algorithms in complex hierarchical embodied systems.

## 1 Introduction

The manual synthesis of a single robot for a single task requires years if not decades of labor-intensive R&D from teams of engineers who rely heavily on expert knowledge, intuition, and hard-earned experience. The automatic synthesis of robots may reveal designs that are different from or beyond what human engineers were previously capable of imagining and would be of great use if robots are to be designed and deployed for many disparate tasks in society. However, automating the creation and improvement of a robot's design is notoriously difficult due to the hierarchical and combinatorial nature of the optimization problem: the objective and variables of the robot's controller optimization depend on the optimizer of the robot's discrete mechanical structure (e.g. its distribution of motors and limbs). Considerable effort has been and is being expended to ameliorate (Zhao et al., 2020; Yuan et al., 2022; Strgar et al., 2024) or obviate (Hejna et al., 2021; Gupta et al., 2022; Matthews et al., 2023) these technical challenges. Nevertheless, automatically designed robots

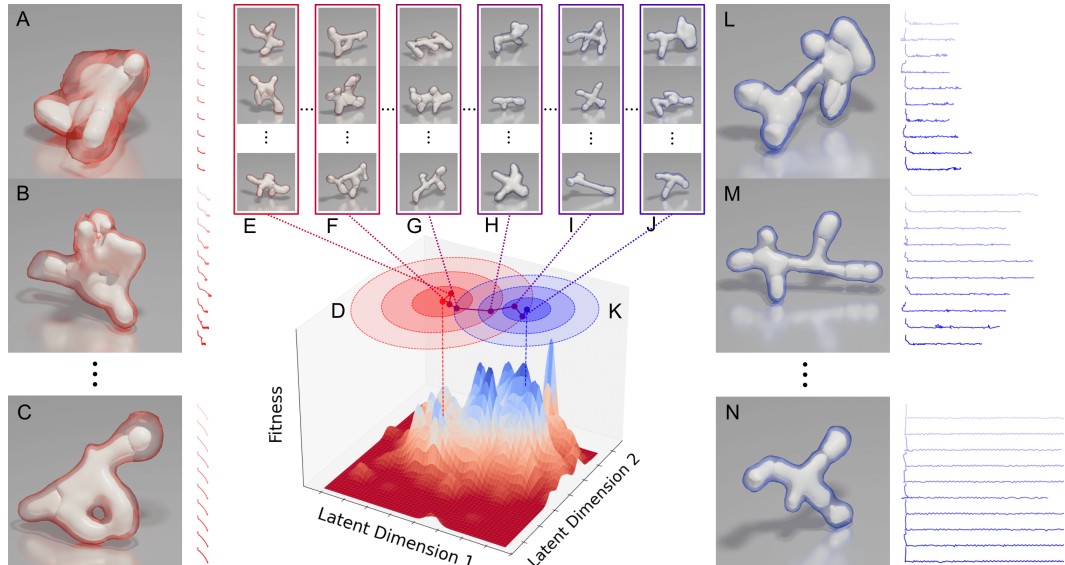

Figure 2: **Designing endoskeletal robots.** A population of 64 robot designs (**A-C**) was sampled from a multivariate normal distribution (**D**) centered about an initially random point in latent space. The behavior each design in the population was optimized in physics simulation for 20 epochs of reinforcement learning under a shared universal controller. The ten red behavioral traces to the right of A-C project the forward locomotion of these randomly generated designs (from 3D to 2D) into the right hand side of the page, at every other epoch of learning. The fitness of a design was based on the best performance it achieved during reinforcement learning. After learning, the mean and covariance matrix of the design distribution were then adapted by evolutionary strategies and a subsequent generation of 64 new designs was sampled from the shifted distribution (**E**). This process was repeated for dozens of generations (**F-J**) yielding an evolved design distribution (**K**) that encodes a population of designs (**L-N**) with much higher fitness (blue behavioral traces).

have yet to evolve beyond primitive machines with exceedingly simple body plans, which can be neatly divided into two opposing groups:

1. fully rigid, jointed bodies (Sims, 1994; Lipson & Pollack, 2000; Auerbach & Bongard, 2014; Zhao et al., 2020; Hejna et al., 2021; Gupta et al., 2021; Yuan et al., 2022); and

2. fully soft, jointless bodies (Hiller & Lipson, 2012; Cheney et al., 2018; Kriegman et al., 2020a; Matthews et al., 2023; Wang et al., 2023b; Li et al., 2024; Strgar et al., 2024).

Here we introduce the de novo optimization of externally soft yet internally rigid, jointed body plans—endoskeletal robots (Fig. 1)—with minimal assumptions about the robots' external and internal structures (Fig. 2). Endoskeletal robots contain bones and joints—the lever systems animals use to their mechanical advantage as they gallop across uneven surfaces, climb up trees, and swing from branch to branch. At the same time, endoskeletal robots contain and are encased by soft deformable tissues—the springs animals use to conform to uneven surfaces, absorb shocks, store and release elastic energy, catapult themselves into the air, and capture complex somatosensory signals about their environment. In order to achieve this confluence of soft and rigid body dynamics within a single robot—without presupposing the topology or the geometry of the robot's bones and tissues—a novel multi-physics voxel-based simulator (Fig. 4) was created and is introduced here alongside a latent endoskeletal embedding and companion generator that sythensizes elastic and rigid voxel cells into freeform tissues and bones, and connects the bones by joints within a functional whole.

Several existing simulators support two-way coupling between rigid and deformable bodies with joint constraints (Shinar et al., 2008; Kim & Pollard, 2011; Li et al., 2020). However they are not naturally amenable to structural optimization because they do not readily expose building blocks (e.g. voxels) to the optimizer; instead, they require a bespoke volumetric mesh to be supplied for each body plan, and the mesh must be carefully designed to remain numerically stable. Jansson & Vergeest (2003) embedded rigid bodies within a regular grid of passive Hookean springs and masses, which provides suitable building blocks of freeform structure, but does not support joint constraints,

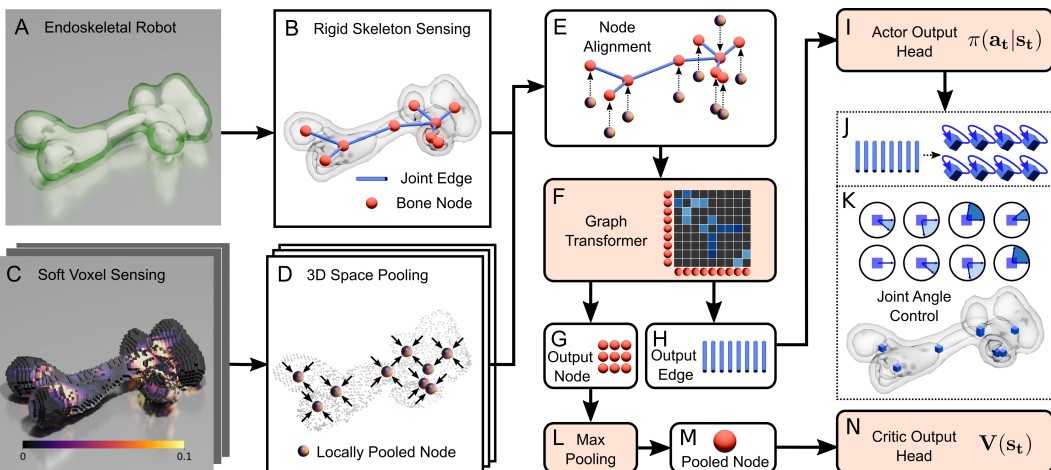

Figure 3: **A universal controller for freeform endoskeletal robots.** The control of an endoskeletal robot (**A**) utilizes a graph data structure with edges that map bone connections within its skeleton, edge features that track the position, orientation and angle of each joint, and node features that track the position, orientation and movement of each bone (**B**). The controller also tracks proprioceptive and mechanosensory input from the position, velocity and strain of the robot's soft tissues (**C**), which is locally pooled into the center of mass of each bone (**D**) thereby transforming the soft tissue's sensory input from voxel space to a graph that aligns with the skeletal sensory graph (**E**). The combined rigid+soft sensory graph is then fed as input to a graph transformer (**F**). The graph transformer distils sensory signatures across the graph into updated node features (**G**), and updated edge features (**H**). The Actor (**I**) takes the updated edge features as input and outputs motor commands (**J**): the target rotation angle for each joint (**K**). The updated node features are pooled by globally by a channel-wise maximum across the node dimension (**L**) to retrieve a graph-level output (**M**), which the Critic (**N**) uses to predict a value function based on the robot's current state information.

actuation, volume preservation under twisting deformations, or the parallelization necessary to scale design to complex bodies composed of large numbers of building blocks. Here, in contrast, we present a massively-parallel simulator that uses a more accurate, twistable model of elasticity (Hiller & Lipson, 2014) as the basis of building blocks of soft tissues which directly attach to rigid building blocks of bones and articulated skeletons with joint constraints and actuation (Baraff, 2001).

Cheney et al. (2013) evolved robots composed of elastic building blocks they called "muscle", "fat" and "bone". The muscles rhythmically pulsed, expanding and contracting in volume; the fat was passive; and the "bone" was equivalent to fat in density and mass but had an order-of-magnitude higher stiffness (Young's modulus). They found that any further increases to bone stiffness prevented the pulsating muscle blocks from sufficiently deforming the robot's jointless body and thus inhibited locomotion. In reality, animal bones are seven orders-of-magnitude stiffer than animal fat (Guimarães et al., 2020), and are pulled by tendons and muscles about joints to rapid generate large yet precise dynamic movements of load bearing limbs. One could simply add joints to this elastic model; however, accurate simulations of elastic elements with higher stiffness (and thus higher resonance frequencies) approaching that of animal bones quickly become prohibitively expensive as ever smaller time steps are required to avoid numerical instability. Moreover, the fully "soft" (entirely non-rigid) and jointless robots evolved by Cheney et al. (2013) also lacked sensors, were composed of no more than a few hundred elastic blocks, and could only move along a perfectly flat surface plane. Here, in contrast, we introduce a new approach that combines hundreds of thousands of both elastically deformable and rigid building blocks into multi-physics bodies that use joints and sensors to navigate complex terrestrial environments.

Following Hiller & Lipson (2010), Compositional Pattern Producing Networks (CPPNs; Stanley (2007)) have been used by many others (Cheney et al., 2013; Auerbach & Bongard, 2014; Cheney et al., 2018; Kriegman et al., 2021; Wang et al., 2023a; Cochevelou et al., 2023) to encode robot designs. CPPNs are a class of feedforward neural networks with spatial inputs (e.g. cartesian coordinates) and hence spatially correlated outputs, which can be used to determine the absence or presence of a cell at each queried spatial coordinate. CPPNs thus provide a simple way to "paint" material

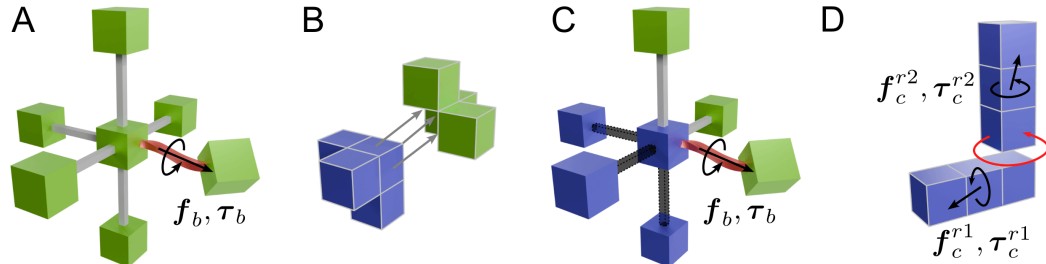

Figure 4: **Simulating endoskeletal robots.** Soft tissues (green masses) were modeled by a grid of Euler–Bernoulli beams (**A**) that may twist and stretch and directly attach to bones (blue masses; **B** and **C**) that follow rigid bodied dynamics with joint constraints (**D**). More details about the simulator can be found in Sect. 2.1 and Appx. B.

into a workspace; however, they struggle to integrate basic structural and functional constraints, such as those which ensure that rigid cells belonging to the same bone form a single contiguous object, that joints are only placed between nearby bones, and that bones remain inside the body. Moreover, one CPPN (its entire architecture) encodes a single design rather than a latent manifold of continuous variables that encode a generative model of robot designs. Here we learn one such continuous representation for endoskeletal robots and demonstrate the learned representation's structural and functional coherence, expressivity, smoothness (Fig. 5) and interpretability (Figs. 13-15).

Prior work encoded robot designs into a continuous latent embedding (Hu et al., 2023); however, the design space (Zhao et al., 2020) was limited by graph grammars to simple body plans containing a small number of jointed rigid links with predefined shapes. These simplifying assumptions allowed for effective model predictive control (Lowrey et al., 2019), which requires an explicit and accurate model of the robot's dynamics, including its interaction with the environment. Here, in contrast, we use a model-free, end-to-end learning approach to design and control externally-soft body plans with highly non-linear dynamics and multi-physics interactions that are difficult to model explicitly.

More specifically, we here train a single reinforcement learning system—a "universal controller" (Fig. 3)—across an evolving population of freeform endoskeletal robots. This is achieved by locally pooling (Peng et al., 2020) each robot's arbitrary arrangement of high-resolution (voxel-level) sensory channels to align with its skeletal graph, which may then be fed as input to a graph transformer (Yun et al., 2019). Graph transformers were previously employed for universal control of many robots with differing physical layouts (Gupta et al., 2022); however, these robots were constrained to bilaterally symmetrical "stick figures" composed of jointed rigid cylinders (Ventrella, 1994). There was no need to learn a 3D volumetric representation of sensor integration as the stick figures were already in a simple graph form with their sensors directly aligned to the edges of their graph. We here evolve embodied agents beyond the rigid stick figures (Yuan et al., 2022; Gupta et al., 2022) and boneless "blobs" (Wang et al., 2023a; Huang et al., 2024) found in recent work with the multiphysics simulation, freeform design and universal control of endoskeletal robots.

## 2 METHODS

In this section we describe how endoskeletal robots were simulated (Sect. 2.1), encoded (Sect. 2.2) and optimized (Sect. 2.3). A symbol table is provided on the very last page of the paper, in Table. 9.

### 2.1 SIMULATING ENDOSKELETAL ROBOTS

We here introduce a novel voxel-based experimental platform that was built from the ground up to support the development of novel representations and algorithms for co-designing the form and function of endoskeletal soft robots. Fig. 4 provides a high level overview of the underlying physics simulator, which enables two-way coupling between freeform soft tissues and freeform rigid bones with joint constraints. In this voxel-based simulator, elastic voxels and rigid voxels are used as building blocks of soft tissues and bones, respectively. Elastic voxels and rigid voxels are subject to their own respective update rules, which are outlined in Appx. B.

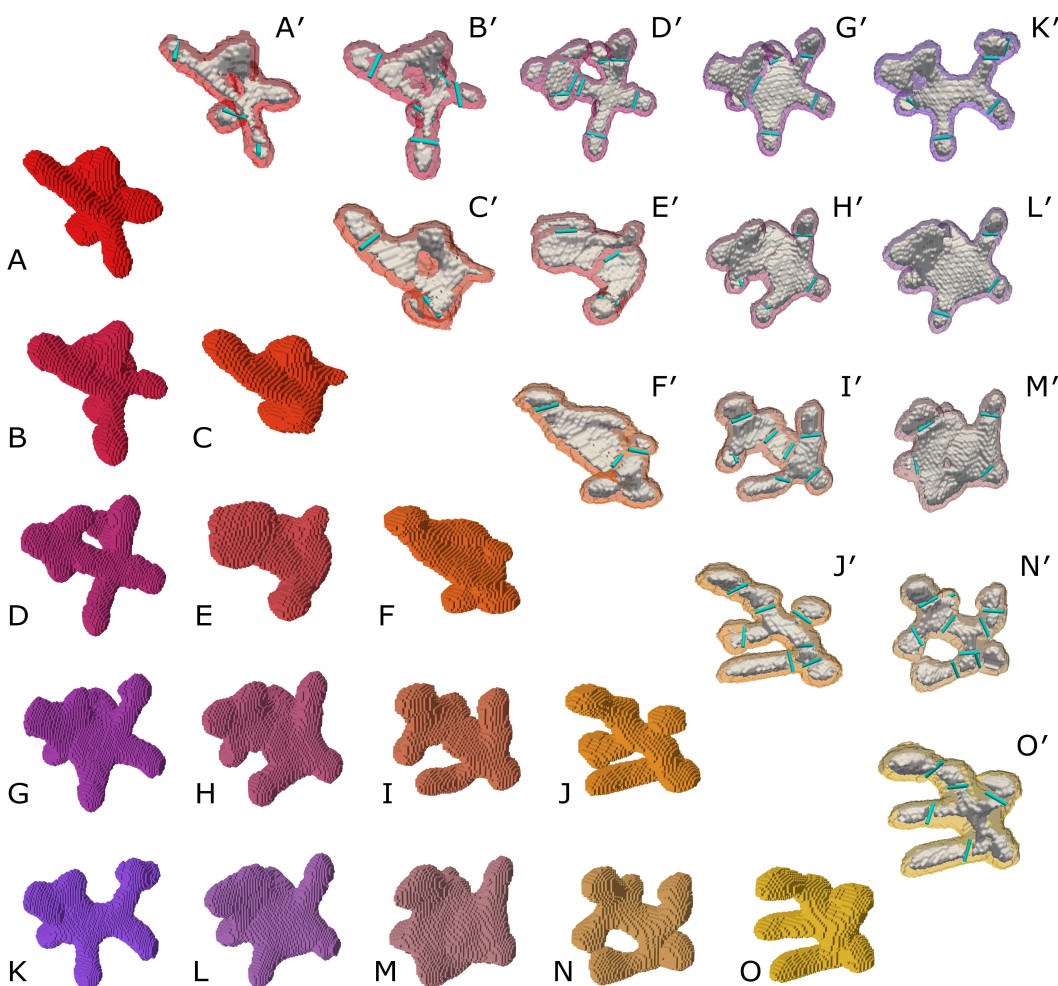

Figure 5: **Interpolating between three points in endoskeletal latent space.** Designs sampled from a 2D slice of the learned latent embedding (**A-O**) and their internal jointed skeletons (**A′-O′**), where A′ reveals the skeleton of A and cyan cylinders show the location of hinge joints between bones in each skeleton. The three corner designs (A,K,O) were drawn from arbitrarily selected latent coordinates and the 12 designs between them sit with equal spacing along a plane of linear interpolation. The visible smoothness, mechanical coherence, and geometric and topological expressiveness of the voxel-based latent space facilitated the co-design of morphology and control.

The most challenging part of building such a simulator was integrating the dynamics of soft and rigid components, which can be summarized by a single equation for all voxels:

$$M^v \ddot{X}^v = F_b^v + F_{\text{ext}}^v + \mathbb{I}(v \in \text{bone})F_c^v, \tag{1}$$

where $v$ is a single voxel, $M^v \in \mathbb{R}^{6 \times 6}$ is the generalized mass matrix combining mass and inertia terms, $\ddot{X}^v \in \mathbb{R}^6$ is the second derivative of generalized position $X^v$ with respect to time $t$, $F_b^v \in \mathbb{R}^6$ and $F_{\text{ext}}^v \in \mathbb{R}^6$ are the generalized force terms including force and torque terms, $F_b^v$ is the effect of all elastic beams (Fig. 4A) and $F_{\text{ext}}^v$ is the effect of all external forces such as gravity and friction. $F_c$ is the generalized constraint force term which is only applied on rigid voxels using the indicator function $\mathbb{I}(v \in \text{bone})$. For more details see Appx. B.

## 2.2 ENCODING ENDOSKELETAL MORPHOLOGY

We began by procedurally generating synthetic training data—examples of valid endoskeletal body plans—in 3D voxel space using multi-star graphs (Fig. 10A), which consist of random number of randomly positioned star nodes like this ✳ with a variable number of arms radiating from their center

points. The radius and length of the bone and the radius of the surrounding soft tissue in each arm were all randomly assigned. Each arm may connect by a joint to an arm of an adjacent star, or remain unconnected. Synthetic data generation pseudocode and hyperparameters can be found in Appx E. The resulting body was voxelized within a $64 \times 64 \times 64$ cartesian grid. Each voxel in the grid was assigned an integer label $i \in [1, K+2]$, where $i = 1$ represents empty space, $i = 2$ represents a soft tissue, and $i > 2$ was used to represent the (at most $K$) independent rigid bones. A one-hot encoding was used to represent the workspace: a tensor with shape $(64, 64, 64, k+2)$.

A variational autoencoder (VAE; Kingma & Welling (2013)) with four blocks of Voxception-ResNet modules (Brock et al., 2016) was then trained to map voxel space into a highly compressed latent distribution consisting of 512 latent dimensions. Prior work that used voxel-based autoencoders considered fixed dataset, which limited their depth and required data augmentation to avoid overfitting (Brock et al., 2016). Because we can generate new synthetic data points on demand, our training data is unlimited, and the depth of our autoencoder—and thus its potential for compress, generalize and capture complex hierarchical features—is not constrained by lack of data.

Each design decoded from the latent space was automatically jointified (as detailed in Fig. 9), and any exposed bone was covered with a thin layer of soft tissue. This tissue patching process was highly targeted; in randomly generated designs, patching was applied to less than 3% of the workspace and covered less than 13% of the robots' bodies. See Appx. F for VAE hyperparameters.

## 2.3 Co-Optimizing Endoskeletal Morphology and Control

**State space.** Endoskeletal robots receive high dimensional mechanosensory data $\mathbf{s}^v = \{(x^v, y^v, z^v, \epsilon^v, \mathbf{v}^v), \ldots \mid v \in \text{soft}\}$ with 3D position $x, y, z$, strain $\epsilon$ and velocity $\mathbf{v}$ from a variable number of soft voxels that are distributed at variable locations in 3D space (Fig. 3C), depending on the size, shape and deformation of its soft tissues. In addition to sensing the movement and distortion of the soft tissues, proprioception of each rigid bone $r$ and joint $j$ between bones is also taken as input. The state of bones can be represented as $\mathbf{s}^r = \{(x^r, y^r, z^r, m^r, \mathbf{q}^r, \mathbf{v}^r, \boldsymbol{\omega}^r), \ldots \mid r \in \text{bone}\}$ where $x, y, z$ is the position of the center of mass of the bone, $m$ is the mass, $\mathbf{q}$ is quaternion representing the orientation, and $\mathbf{v}$ and $\boldsymbol{\omega}$ are the linear velocity and angular velocity. The state of joints can be represented as $\mathbf{s}^j = \{(\mathbf{h}^j, \mathbf{d}_1^j, \mathbf{d}_2^j, \theta^j, \theta_{\min}^j, \theta_{\max}^j), \ldots \mid j \in \text{joints}\}$ where $\mathbf{h}$ is the hinge axis direction of joint, $\mathbf{d}_1$ is the vector from the joint position to the center of mass of the first connected bone, $\mathbf{d}_2$ is the vector from the joint position to the center of mass of the second connected bone, $\theta^j$ is the current joint angle, and $\theta_{\min}^j$ and $\theta_{\max}^j$ are the minimum and maximum joint angle limits, respectively. Soft voxel sensing is locally pooled into nodes at the center of mass of each bone (Fig. 3D) thus aligning soft tissue and skeletal inputs (Fig. 3E) and promoting "reflex partitioning" (Windhorst et al., 1989) whereby motors are affected more by local sensors than distant sensors.

**Action space.** The action space consists of joint angles, which control the movements and articulation of the robot's internal skeleton. We use a uniform discrete angle action space $\mathcal{A} = \{\text{-1.4 rad, -0.7 rad, 0 rad, 0.7 rad, 1.4 rad}\}$ for all joints. We selected 1.4 rad as the limit for joint rotation range to prevent extremely large unrealistic deformation of the soft materials surrounding the joints (Bonet & Wood, 2008). We found that a discrete action space greatly simplified and stabilized policy training compared to an otherwise equivalent continuous action space (Fig. 16).

**Reward.** Reward was based on net displacement of the robot across 100 pairs of observations and actions sampled at 10Hz during a simulation episode of 10 seconds. Rewards were generated per episode in the following way:

$$\mathbf{u}_t = (x_{t-1} - x_0, y_{t-1} - y_0) \tag{2}$$

$$\hat{\mathbf{u}}_t = \frac{\mathbf{u}_t}{||\mathbf{u}_t||} \tag{3}$$

$$\mathbf{v}_t = (x_t - x_{t-1}, y_t - y_{t-1}) \tag{4}$$

$$r_t = \begin{cases} -P_{\text{large}}, & \text{if } ||\mathbf{v}_t|| < \delta \\ \max(\hat{\mathbf{u}}_t \cdot \mathbf{v}_t, -P_{\text{small}}), & \text{otherwise} \end{cases} \tag{5}$$

where $x_t$ and $y_t$ are the x and y position of the center of mass of the robot at time step $t$, $r_t$ the reward received by the robot, $\delta$ is a small threshold value for detecting stationary, and $\mathbf{u}_t$ is the history movement vector up to the previous frame projected to the xy plane. If the norm of $\mathbf{u}_t$ is

0, we replace $\hat{\mathbf{u}}_t$ with vector $(1, 0)$ pointing in the positive X direction. We apply a large penalty if robot is static and clip a negative reward to a smaller penalty if the robot is moving but in an undesired direction. Eq. 5 encourages the robot to maintain consistent forward motion while penalizing stagnation or drastic directional changes, which was found to be more robust for identifying designs with high locomotive ability compared to using a naive reward based on moving distance in a fixed direction, such as $r_t = x_t - x_{t-1}$. This is because the naive reward may discard designs with high locomotive ability in the wrong direction. In some experiments the reward function was augmented to encourage robots to lift themselves above the surface plane using legs.

**Policy.** We employed Proximal Policy Optimization (PPO; Schulman et al. (2017)) to train a single universal controller for an evolving population of 64 endoskeletal robots. A clone of each design in the population was created, yielding a batch of 128 designs. This last step is intended to broaden the evaluation of each design and so reduce the likelihood of erroneously discarding a good design. At each time step $t$, the policy's observation $\mathbf{s}_t = (\mathbf{s}_t^v, \mathbf{s}_t^j, \mathbf{s}_t^r)$ consists of soft voxel observations $\mathbf{s}_t^v$, joint observations $\mathbf{s}_t^j$, and rigid bone observations $\mathbf{s}_t^r$. Given these observations as input, the policy returns actions and value estimates according to the following equations:

$$\mathbf{s}_t^l = \text{SpatialPool}(\mathbf{s}_t^v) \tag{6}$$

$$\mathbf{n}_t = [\mathbf{s}_t^r, \mathbf{s}_t^l] \tag{7}$$

$$(\tilde{\mathbf{n}}_t, \tilde{\mathbf{e}}_t) = \text{GraphTransformer}(\mathbf{n}_t, \mathbf{s}_t^j) \tag{8}$$

$$\mathbf{n}_t^* = \text{MaxPool}(\tilde{\mathbf{n}}_t) \tag{9}$$

$$V(\mathbf{s}_t) = \text{ResNet}_V(\mathbf{n}_t^*) \tag{10}$$

$$\mathbf{z}_{t,j} = \text{ResNet}_\pi(\tilde{\mathbf{e}}_{t,j}) \tag{11}$$

$$\pi(a_{t,j}|\mathbf{s}_t) = \text{Softmax}(\mathbf{z}_{t,j}) \tag{12}$$

$$\pi(a_t|\mathbf{s}_t) = \prod_j \pi(a_{t,j}|\mathbf{s}_t) \tag{13}$$

Soft voxel observations $\mathbf{s}_t^v$ are locally pooled (Peng et al., 2020) into a $64^3$ grid using the robot center of mass as the grid origin. The convoluted feature grid is then queried at the center of masses of each rigid bone, forming locally pooled node features $\mathbf{s}_t^l$ (Fig. 3D). This operation aligns soft tissue sensory inputs with skeletal sensing. The pooled features are then concatenated with the rigid bone observations $\mathbf{s}_t^r$ to form the node features $\mathbf{n}_t$, as shown in Fig. 3E.

A graph transformer (Shi et al., 2021) processes the robot's topology graph, taking the node features $\mathbf{n}_t$ and edge features $\mathbf{s}_t^j$ (joint observations) as input, and outputting processed node features $\tilde{\mathbf{n}}_t$ and processed edge features $\tilde{\mathbf{e}}_t$. The processed edge features are produced by concatenating processed node features across the two nodes connected by each edge. By max pooling (Fig. 3L) over the processed node features $\tilde{\mathbf{n}}_t$ we obtain a global feature vector $\mathbf{n}_t^*$ (Fig. 3M), which summarizes the overall state of the robot. The Critic (Fig. 3N) uses $\text{ResNet}_V$ to take the pooled node feature $\mathbf{n}_t^*$ and output the value function $V(\mathbf{s}_t)$. The Actor (Fig. 3I) processes edge features $\tilde{\mathbf{e}}_{t,j}$, corresponding to each joint $j$, and passes them through $\text{ResNet}_\pi$ to compute the action logits $\mathbf{z}_{t,j}$ which define the action distribution for joint $j$. By computing the softmax over the logits $\mathbf{z}_{t,j}$, probabilities are obtained over the discrete action space $\mathcal{A}$. The overall policy $\pi(a_t|\mathbf{s}_t)$ was thus defined as the product of the action distributions over all joints, assuming independence between joints.

By utilizing a graph transformer to process the robot's topology, the policy effectively learns to condition on both the sensory input and the morphology of a robot. This architecture allows the actor to generate joint-specific actions while the critic evaluates the discounted score $V(\mathbf{s}_t) = \mathbb{E}_\pi\left[\sum_{t=0}^T \gamma^t r_t\right]$.

**Fitness.** The aggregate behavior of a robot across an entire simulation episode was evaluated against a fitness function. After 20 epochs of learning, each design in the current population was assigned a fitness score equal to the peak fitness it achieved across 40 episodes of simulation.

**Evolutionary design.** The population of designs fed to the universal controller was optimized by covariance matrix evolutionary strategies (CMA-ES; Hansen & Ostermeier (2001)). Briefly, a multivariate normal distribution of designs is sampled from the latent space and the mean vector is "pulled" toward the latent coordinates of sampled designs with the highest fitness and the covariance matrix is adapted to stretch along the most promising dimensions and shrink within others.

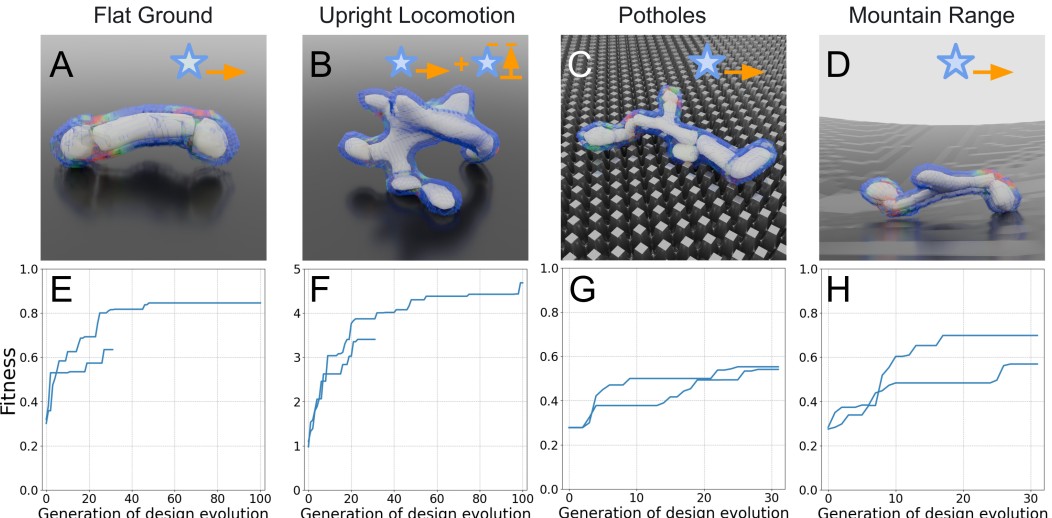

Figure 6: **Task environments.** We considered four task environments: Flat Ground (**A**), Upright Locomotion (on flat ground; **B**), Potholes (**C**) and Mountain Range (**D**). In each one, two independent evolutionary trials were conducted, and the peak fitness achieved by each design was averaged across the population before plotting the cumulative max (**E-H**). The best design is shown for each environment (A-D). Rewarding net displacement over flat ground (A,E) results in the evolution of snakes (A). We ran optimization for extra time (100 generations) to see if legs would emerge, but evolution failed to escape the local optima of snakes. Adding a second component to the reward function that tracks the proportion of body voxels that fall below a prespecified height threshold during behavior (B,F) resulted in legged locomotion (B), and continued to innovate when provided extra time. Increasing the difficultly of the terrain also promoted morphological diversity (C,D).

Because CMA-ES uses ranked fitness scores instead of raw fitness scores to update the mean design vector, undesirable designs can be assigned a large negative rank without interfering with the stability of the optimizer. Body plans that have less than two joints or less than 20% bone, were deemed "invalid designs" and assigned a large negative reward as described below in Sect. 2.3. Less than 15% of the randomly generated robots were invalid under this definition of validity.

## 3 RESULTS

In this section, we present our learned endoskeletal latent embedding and its use in body-brain co-design across four different task environments. We also perform a series of control experiments that isolate the effect of morphological evolution in order to determine its role in achieving high performing endoskeletal robots.

### 3.1 ENDOSKELETAL LATENT SPACE

The learned latent space is smooth and expressive (Fig. 5) and was found to contain tunable genes that encode specific morphological traits, such as body height, length, width, volume and bone count (Figs. 13-15). The trained VAE is able to encode and decode previously unseen training data points with high accuracy (Fig. 10) while generalizing beyond the training data: Most designs sampled from the latent space look very different from the multi-star graphs (see e.g. Fig. 10A) used to train the VAE. For example, skeletal voids (i.e. holes that pass through individual bones) were not present in the generated synthetic training data set but they are present in the bones of decoded designs from randomly-sampled latent codes (Fig. 5A′).

### 3.2 TASK ENVIRONMENTS

**Flat Ground.** On perfectly flat ground, selection for forward locomotion resulted in the evolution of snake-like body plans with two joints (Fig. 6A) that are able to move at high speeds.

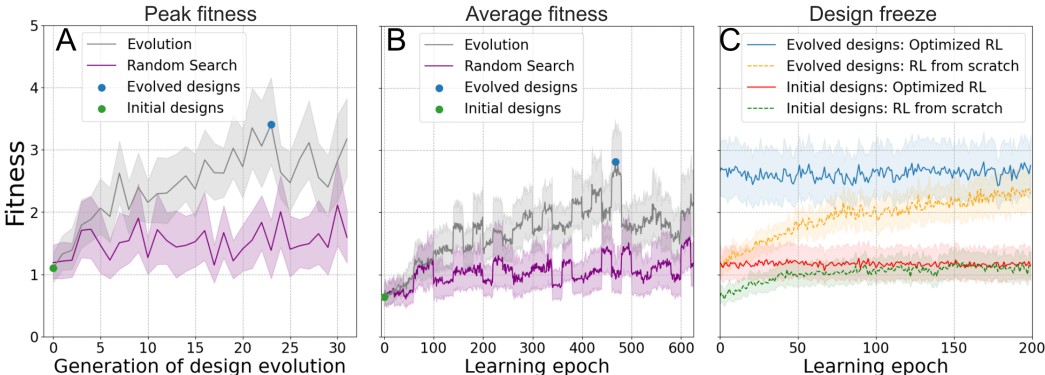

Figure 7: **Evolution of learning.** A population of 64 designs was iteratively drawn from an evolving design distribution and used by reinforcement learning to obtain a universal controller. The peak fitness achieved by each design across 20 epochs of RL is plotted (with 95% bootstrapped confidence intervals) at each generation of design evolution in the Upright Locomotion task environment (**A**). The dark gray line in A is mean peak fitness across the 64 designs in the population; the shaded region is the CI of the mean. Similarly, the average fitness achieved by each design across its simulation episodes is shown at each learning epoch (in **B**). An otherwise equivalent experiment with random morphological search instead of evolution strategies was also performed (purple curves in A and B). After 30 generations of evolution (600 total epochs of RL), the 64 designs from the best evolved population (blue dot; gen 23) were compared to the 64 designs from initial randomly-generated population (green dot; gen 0). To do so, the designs within each population were frozen and their behavior was re-optimized (**C**) starting with the optimized universal controller from gen 23 (Optimized RL). A second comparison was then made in which a new universal controller re-learned for each population separately, from scratch (RL from scratch). In both comparisons, the learning conditions were equivalent and the only differences between the populations were morphological. And in both comparisons the evolved designs easily outperformed the initial designs. This indicates that design evolution was necessary: it pushed search into regions of the latent space containing better morphologies with higher fitness potential that better support universal controller training.

**Upright Locomotion.** Given the impressive speed of snakes on flat ground, a second term was added to the reward function to promote body plans that exhibit upright legged locomotion. To do so, the proportion of body voxels that fell within 5 cm of the surface plane during behavior was subtracted from the original reward at each step of learning, and the original fitness was divided by the mean proportion of voxels that fell below this threshold across the full simulation episode. This resulted in less serpentine body plans with more joints (Fig. 12). The best evolved design is much more complex, and uses its legs to lift its body above the ground during locomotion (Fig. 6B).

**Potholes.** Next, we explored a more challenging terrain. In the Potholes environment, there are regularly-spaced depressions ("potholes") along the surface plane. The depth, width, and spacing of the potholes can be easily adjusted in the provided code (indeed any 2D height matrix can be supplied to change the terrain). With relatively small, tightly-spaced potholes, the best evolved designs (Fig. 6C) are longer and more complex than the snakes that dominated flat ground under the same fitness function (Fig. 6A). A common adaptation that evolved for navigating potholes was broad feet, which the robots used to push off and pull against the vertical walls inside the potholes.

**Mountain Range.** In our fourth and final environment, robots must climb up a slope in order to move away from the origin. In this mountain environment, the best evolved designs (Fig. 6D) were also slightly more complex than the flatland snakes.

### 3.3 THE NECESSITY OF MORPHOLOGICAL EVOLUTION

Across all four task environments, evolved design populations significantly outperform the randomly-generated initial populations of designs from the first generation of evolution. This could be due to improvements in the universal controller or better body plans, or both. In order to isolate the effects of morphological evolution and policy training, we performed the following three control experiments using the Upright Locomotion task environment.

In the first experiment, we extracted the design population that achieved the highest mean reward during evolution (Evolved designs in Fig. 7B), as well as the RL model checkpoint at this point (Optimized RL in Fig. 7C). We also extracted the initial designs that were randomly generated at the very beginning of evolution (gen 0). The initial designs (Fig. 11) were frozen and the controller was re-optimized using this population of frozen designs for 100 epochs starting from optimized RL checkpoint. In a parallel independent experiment, the evolved designs (Fig. 12) were likewise frozen and the controller was re-optimized using the frozen evolved population of designs for 100 epochs starting from optimized RL checkpoint. The same procedure was repeated, this time retraining the control policy from scratch, for both frozen design populations separately. In both cases—starting policy training from scratch or from the RL checkpoint—the frozen evolved designs significantly outperformed the frozen initial designs (Fig. 7C). In both frozen design populations, the universal controller from the checkpoint immediately exhibits the best performance achieved after retraining from scratch. This shows that RL does not suffer catastrophic forgetting: the policy does not forget how to control the initial designs as the population evolves to contain different, better designs.

These results suggest that morphological evolution yields better designs but they do not prove that evolution is necessary. It could be the case that good designs will simply arise by random mutations alone, without cumulative selection across discrete generations. As a final control experiment we tested this hypothesis by replacing evolutionary strategies with random search (purple curve in Fig. 7A,B). Random morphological search performed significantly worse than evolutionary strategies. This suggests that morphological evolution (mutation *and* selection) is indeed necessary.

## 4 DISCUSSION

In this paper we introduced the computational design of freeform endoskeletal robots. The open-ended nature of the design problem allowed us to procedurally generate a never-ending synthetic data flow of endoskeletal training examples for representation learning, which in turn allowed for effective learning of a much deeper voxel-based autoencoder than those learned in prior work with fixed datasets (Brock et al., 2016). This resulted in a very smooth and highly expressive latent embedding of design space (Fig. 5), which we employed as a genotype space (and the decoder as the genotype-phenotype mapping) for morphological evolution.

We found that a universal controller could be simultaneously obtained by reinforcement learning during morphological evolution despite the wide range of endoskeletal geometries and topologies that emerged within the evolving population. We observed across several independent experimental trials that morphological evolution tended to push search into promising regions of latent space consisting of high performing body plans that facilitated policy training. For flat ground, this resulted in the evolution of simple two-jointed snakes (Fig. 6A), which in hindsight makes intuitive sense: such bodies are easier to control. But, in the other three task environments we tested, many other, more complex solutions evolved, including legged locomotion (Fig. 6B). This suggests that other researchers may leverage the open-endedness of this computational design platform to identify which physical environments and fitness pressures lead to the emergence of particular abilities and the embodied structures required to support those abilities

Others may use our simulator as an easy-to-integrate software library for benchmarking their own voxel-based representations and learning algorithms in the four task environments released here, or by creating their own terrain maps and reward functions based on these examples. To this end, we also provide an example of object manipulation ("dribbling a football"; Fig. 8), which we hope inspires others to create their own objects and build more, and more intricate, virtual worlds.

However, there were key limitations of the present work that may be overcome by future work. First, because skeletons were assumed to be inside of each robot, rigid body collisions were not modeled, and thus external rigid horns, claws, shells, etc. were not possible. Also, fluids were not modeled, and thus the simulator was restricted to terrestrial environments and land based behaviors. But the most important limitation of this paper is that simulated designs were not realized as physical robots. Recent advances in volumetric 3D printing (Darkes-Burkey & Shepherd, 2024) are beginning to enable the fabrication of endoskeletal bodies that flex their muscles but are not yet capable of locomotion. This suggests that artificial endoskeletal systems will soon move through our world, opening the way to evolved endoskeletal robots that begin to approach the sophisticated—and perhaps eventually cognitive behavior of evolved endoskeletal organisms.

ACKNOWLEDGMENTS

This research was supported by NSF award FRR-2331581, Schmidt Sciences AI2050 grant G-22-64506, Templeton World Charity Foundation award no. 20650, and the Berggruen Institute.

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

# A  SUPPLEMENTAL FIGURES

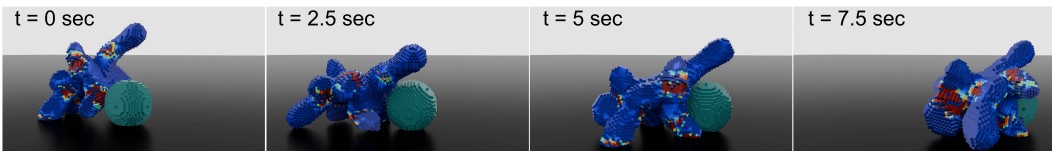

Figure 8: **Object manipulation.** One of the designs that evolved for Upright Locomotion performs a new task: it pushes an object forward. This design was among several others that we sampled from the evolving population analyzed in Fig. 7, placed behind a soft sphere (teal), and actuated using the universal controller from the Optimized RL checkpoint in Fig. 7C. Videos of the resulting behaviors and the source code necessary to reproduce our results can be found on our project page. https://endoskeletal.github.io.

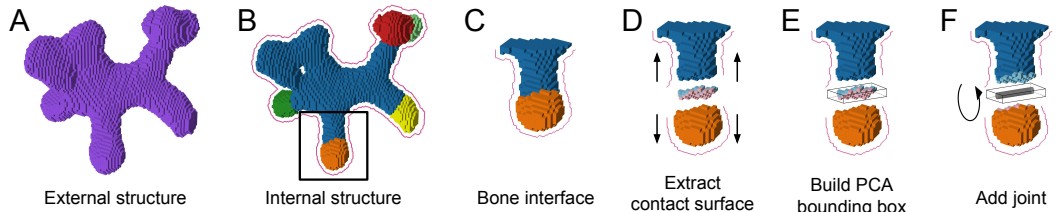

Figure 9: **Jointification.** The learned latent space encodes a generative model of an endoskeletal body plan: a external soft tissue geometry (**A**) and an internally segmented skeleton (**B**). The latent space also implicitly encodes the location and orientation of each joint in the skeleton: at the interface of each pair of bones in voxel space (**C**). In order to simulate a joint, voxels along the contact surface of two opposing bones are extracted (**D**) and Principal Component Analysis is used to construct an oriented bounded box around them (**E**). A hinge joint is then positioned at the center of mass of the contact surface with a direction along with the longest axis of the bounding box (**F**).

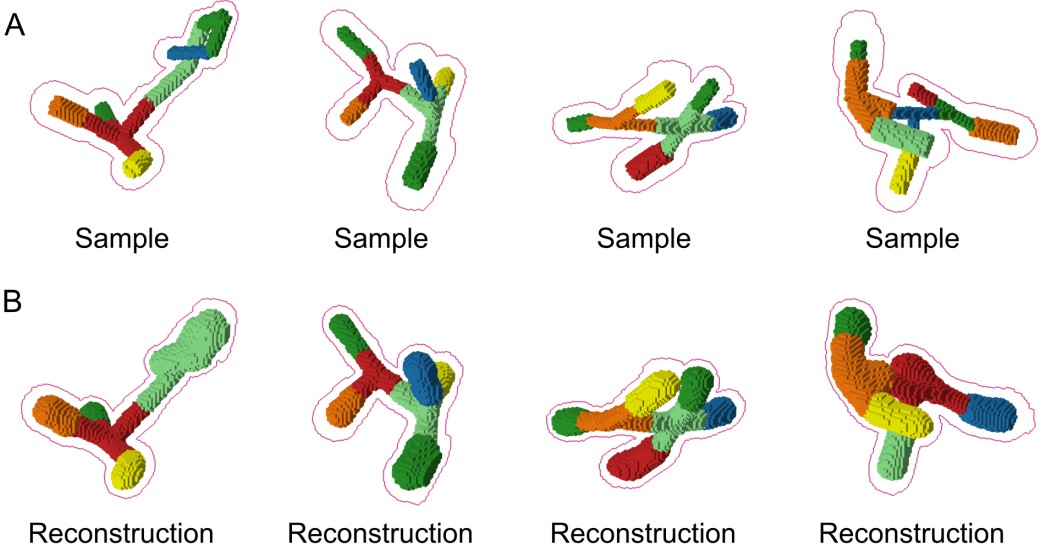

Figure 10: Examples of encoding previously unseen sythentic training data (**A**) and reconstructing each one (**B**) using the fully optimized VAE. Each column displays a sample/reconstruction pair.

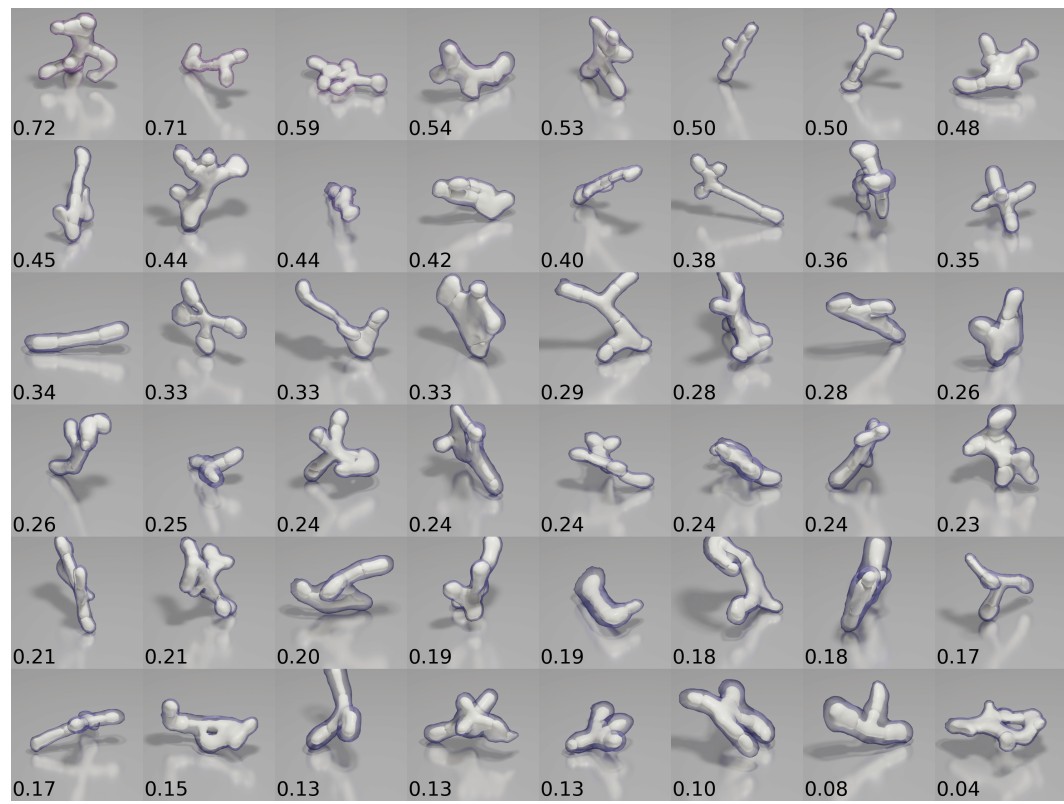

Figure 11: **Ancestors.** Part of a random initial population, sorted by net displacement (in meters).

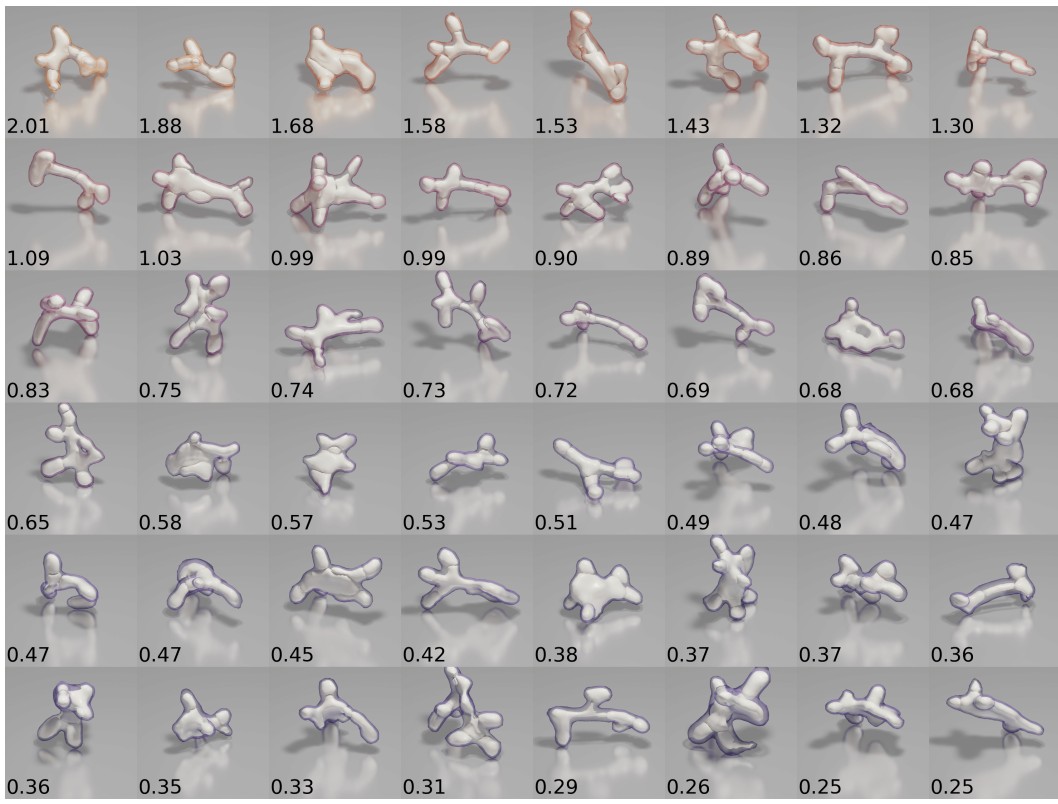

Figure 12: **Descendants.** Part of the evolved population that descended from the initial population shown above in Fig. 11 for Upright Locomotion. Under each design, its net displacement in meters.

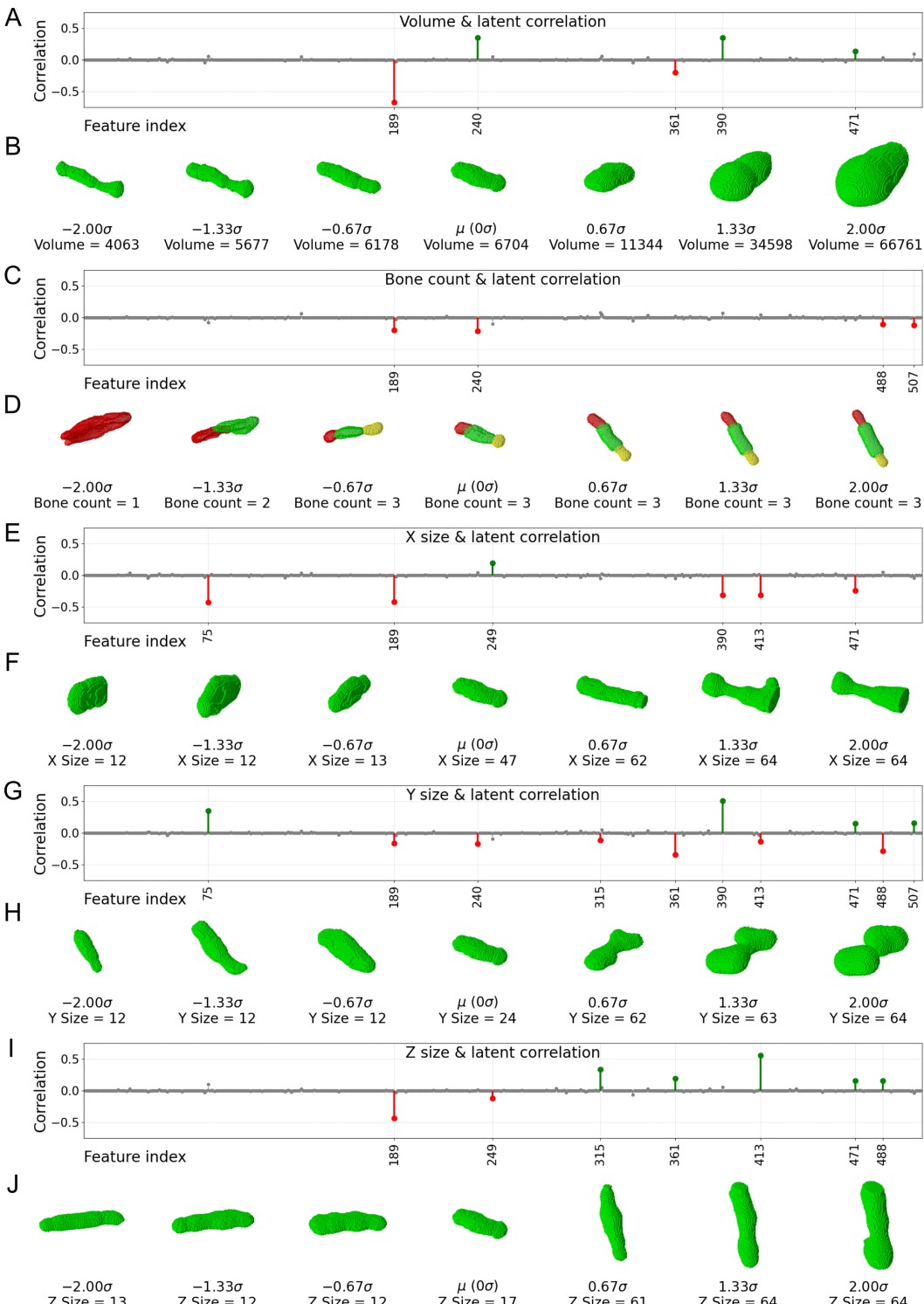

Figure 13: **Screening for latent genes.** We measured the Pearson correlation coefficient ($\rho$) of the 512 latent features and the body volume of one million randomly sampled designs (**A**). The set of latent features that are highly correlated with volume ($|\rho| > 0.1$) were extracted as "the gene for volume", the effect of which can be increased or decreased on demand (**B**). The effect of tuning the gene for volume $\pm 2\sigma$ is shown for the design decoded from the sample mean of the latent space ($\boldsymbol{\mu}$). The same procedure was repeated to identify and modulate genes for bone count (**C, D**), length (**E, F**), width (**G, H**) and height (**I, J**).

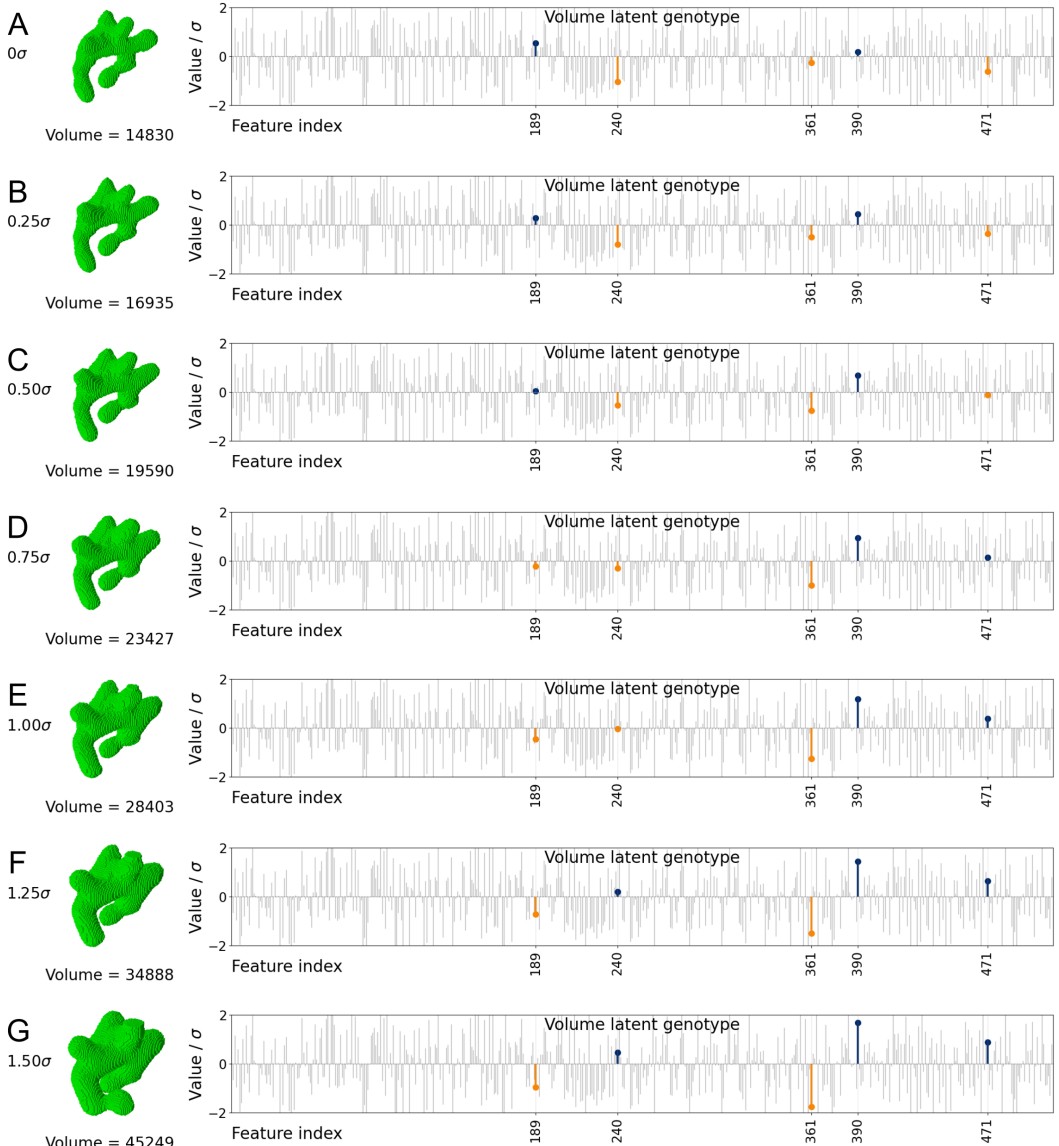

Figure 14: **Gene editing.** The best design that evolved for Upright Locomotion was extracted (**A**) and the set of latent features found to be individually correlated with body volume—the gene for volume—was incrementally increased and the volume of the resulting design was measured (**B**-**G**). The entire latent genome is shown to the right of each decoded design, with features outside the volume gene grayed out. Positively correlated features (240, 390 and 471) were amplified, and negatively correlated features (189 and 361) were reversed by $0.25\sigma$ in each row, yielding progressively thicker body parts within the evolved body shape.

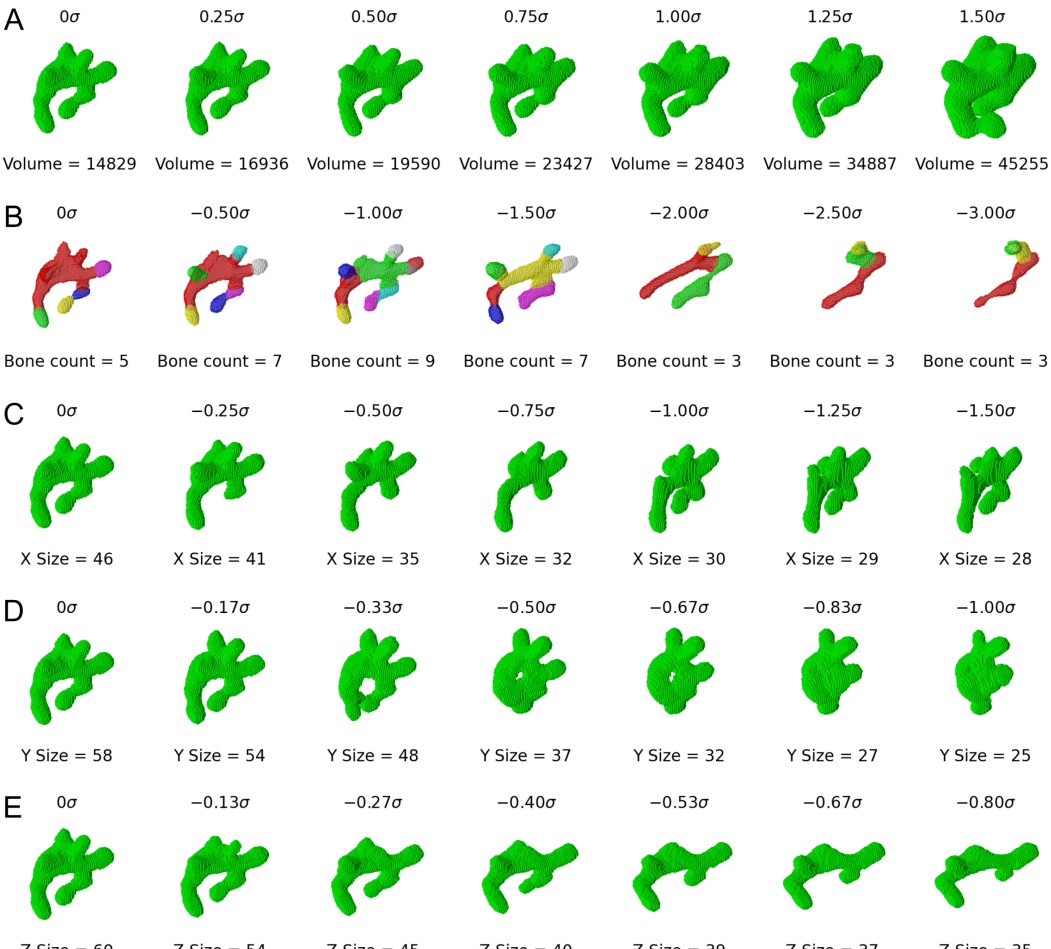

Figure 15: **Programmable morphology.** The best design that evolved for Upright Locomotion was extracted (from Fig. 6B,F) and the found genes for volume (**A**), bone count (**B**), length (**C**), width (**D**) and height (**E**) were individually modulated to demonstrate the precision with which evolved morphologies can be manually redesigned through "genetic engineering". Some genes afford greater control over a specific morphological trait than others; however, this may be attributed to the simplicity of the procedure employed to screen for genes, which only captures pairwise linear associations between a manually-specified morphological trait and an individual latent feature.

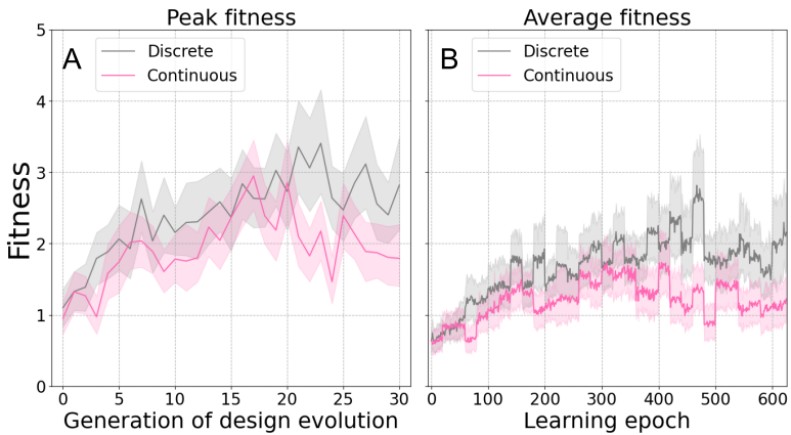

Figure 16: **Discrete and continuous action spaces.** Peak (**A**) and average (**B**) fitness is plotted for the Upright Locomotion task, as in Fig. 7A and B. Two kinds of action spaces were compared: discrete (gray) and continuous (pink). The discrete action space is {-1.4 rad, -0.7 rad, 0 rad, 0.7 rad, 1.4 rad} and the continuous action space is [-1.4 rad, 1.4 rad], for all joints. In both panels, each line is a mean across the 64 designs in the population and the shaded regions are a 95% bootstrapped CI of the mean. Although both continuous and discrete action spaces are viable, a discrete set of actions greatly simplified and stabilized universal policy training under the tested conditions.

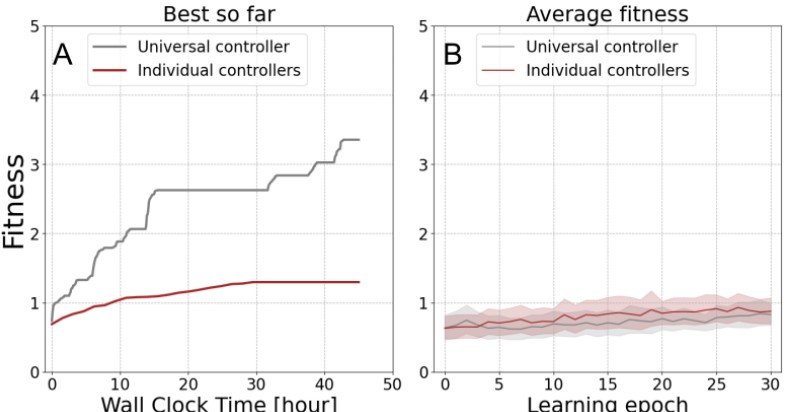

Figure 17: **Universal and individual controllers.** An otherwise equivalent experiment was conducted in which the universal controller shared by all 64 designs in the population was replaced with a bespoke individual controller for each design. As in Fig. 6, the peak fitness achieved by each design was averaged across the population before plotting the cumulative max (**A**). This statistic, which we term "best so far", captures the progress of the entire population. Training 64 independent policies for 30 epochs required 44.8 hours on 4 NVIDIA H100 SXM GPUs. During the same time frame on the same hardware the universal controller can be trained for 451 epochs and thereby achieve much higher fitness. Over the first 30 epochs of learning, the performance of the universal controller was not statistically different from that of the individual controllers (**B**), and since design evolution was only advanced a single generation, the designs under universal control were visually similar to those with independent controllers, with both populations closely resembling the random initial population (Fig. 11).

## B   SIMULATION

In this section we describe the underyling soft and rigid body physics behind our endoskeletal robot simulator. We will use the symbol $\boldsymbol{\tau}$ for both moment and torque to avoid confusion. Subscripts are used to indicate the source of an effect and superscripts indicate the target of an effect.

### B.1   SOFT VOXELS

Following Hiller & Lipson (2014), soft voxels are defined as a point of mass with rotational inertia in space. Each surface of a soft voxel may connect by a Euler–Bernoulli beam with zero mass to an adjacent soft voxel (Fig. 4A) or rigid voxel (Fig. 4B,C). Each beam exerts moment $\boldsymbol{\tau}_b$ and force $\boldsymbol{f}_b$ onto the corresponding connected voxel when deformed from expansion, compression or twisting. Although Euler–Bernoulli beams are more complex to compute than Hookean springs, the additional torsional moment $\boldsymbol{\tau}_b$ contributes to preserving volume when the structure is under external torsional stress. Since endoskeletal robots embed joints between rigid bones inside soft tissue and produce large deformations, preserving volume prevents generating invalid force and moment values when soft tissues are radically compressed and thereby stabilizes simulation.

The moment $\boldsymbol{\tau}_b$ and force $\boldsymbol{f}_b$ generated by beams can be computed from position $\mathbf{x} = (x, y, z)$ and rotation $\boldsymbol{\theta}$ of voxels $v_1$ and $v_2$ attached on both ends using a constant stiffness matrix $K$:

$$\begin{bmatrix} \boldsymbol{f}_b^{v1} \\ \boldsymbol{\tau}_b^{v1} \\ \boldsymbol{f}_b^{v2} \\ \boldsymbol{\tau}_b^{v2} \end{bmatrix} = [K] \begin{bmatrix} \mathbf{x}^{v1} \\ \boldsymbol{\theta}^{v1} \\ \mathbf{x}^{v2} \\ \boldsymbol{\theta}^{v2} \end{bmatrix} \tag{14}$$

Following Newton's second law, we can express the equation of motion for soft voxels as:

$$m^v \ddot{\mathbf{x}}^v = \sum \boldsymbol{f}_b^v + \boldsymbol{f}_{\text{ext}}^v$$
$$I^v \ddot{\boldsymbol{\theta}}^v = \sum \boldsymbol{\tau}_b^v + \boldsymbol{\tau}_{\text{ext}}^v \tag{15}$$

where $f_{\text{ext}}^v$ and $\tau_{\text{ext}}^v$ represent external force and moment caused by gravity, friction, and contacting force etc. For simplicity, we will define generalized mass, position and force representations $M^v$, $X^v$ and $F^v$ and rewrite Eq. 15 as:

$$M^v \ddot{X}^v = \sum F_b^v + F_{\text{ext}}^v \quad \text{with} \quad X^v = \begin{bmatrix} \mathbf{x}^v \\ \boldsymbol{\theta}^v \end{bmatrix}, F^v = \begin{bmatrix} \boldsymbol{f}^v \\ \boldsymbol{\tau}^v \end{bmatrix}, M^v = \begin{bmatrix} m^v E & 0 \\ 0 & I^v \end{bmatrix} \tag{16}$$

where $E$ is the identity matrix.

### B.2   RIGID VOXELS

Rigid voxels are affected by the beams of connected soft voxels (Fig. 4C), generalized external forces $F_{\text{ext}}^v$, and additional generalized constraint forces $F_c^v$, as from joints (Fig. 4D). Since rigid voxels belonging to the same rigid bone $r$ do not have relative motion, we compute the generalized position $X^r$ of the compound rigid bone $r$ and infer $X^v$. As each rigid bone will be connected by many beams to its surrounding soft voxels, we will use $F_b^r$ as the total generalized force exerted by all beams to simplify the representation:

$$M^r \ddot{X}^r = F_b^r + F_{\text{ext}}^r + F_c^r \tag{17}$$

For $F_b^r$ and $F_{\text{ext}}^r$, torque is also generated by forces applied to the center of component rigid voxels. Thus:

$$F_b^r = \sum_{v \in r, b \in v} \begin{bmatrix} \boldsymbol{f}_b^v \\ \boldsymbol{\tau}_b^v + (\mathbf{x}^v - \mathbf{x}^r) \times \boldsymbol{f}_b^v \end{bmatrix} \quad \text{and} \quad F_{\text{ext}}^r = \sum_{v \in r} \begin{bmatrix} \boldsymbol{f}_{\text{ext}}^v \\ \boldsymbol{\tau}_{\text{ext}}^v + (\mathbf{x}^v - \mathbf{x}^r) \times \boldsymbol{f}_{\text{ext}}^v \end{bmatrix} \tag{18}$$

where $x_r$ represents the center of mass of the corresponding rigid bone $r$.

Contacts were modeled between voxel pairs and contact forces were thus aligned to the center of colliding voxels instead of the center of mass of the rigid bones. To prevent generating a false torque term in $F_{\text{ext}}^v$, the contact force does not contribute to the torque.

Finally, constrained dynamics were used to compute $F_c^r$ as follows. Constraints were defined as functions between two rigid bones $r_1$ and $r_2$:

$$c(X^{r_1}, X^{r_2}) \geq 0 \tag{19}$$

For convenience, we now remove superscripts and use concatenated $X$, $F$, and block-diagonalized $M$ to represent attributes of both bones. By taking the time derivative of constraint function $c$ we obtain the Jacobian matrix $J$ which maps global space to constraint space:

$$\dot{c}(X) = J\dot{X} \tag{20}$$

Introducing Lagrange multiplier $\lambda$, which physically represents applied constraint forces, and we can rewrite Eq. 17 as:

$$M\ddot{X} = F + J^T\lambda \quad \text{with} \quad F = F_b + F_{\text{ext}} \tag{21}$$

By substituting Eq. 21 into Eq. 20 we get:

$$\dot{c}(X) = J\dot{X_{\text{old}}} + JM^{-1}(F + J^T\lambda)\Delta t \tag{22}$$

where $\dot{X_{\text{old}}}$ is speed from the last time step; this derivative needs to be non-negative if we wish the constraint to be satisfied at last time point. We also add a bias factor (Baumgarte, 1972) to correct constraint errors incrementally:

$$\dot{c}(X) = J\dot{X_{\text{old}}} + JM^{-1}(F + J^T\lambda)\Delta t + \frac{\beta}{\Delta t}c_{\text{old}}(X) \geq 0 \tag{23}$$

where $\beta$ is a constant that controls the constraint error correction rate in each time step.

Reordering terms in Eq. 23 and we obtain equation system in the form of a linear complementarity problem (LCP) and then solve $\lambda$ using the projected Gauss-Seidel (PGS) algorithm (Erleben, 2013):

$$A = JM^{-1}J^T, \tag{24}$$

$$w = J\dot{X_{\text{old}}} + JM^{-1}F\Delta t + \frac{\beta}{\Delta t}c_{\text{old}}(X), \tag{25}$$

$$A\lambda + w \geq 0 \tag{26}$$

$$\lambda \geq 0 \tag{27}$$

$$\lambda \perp A\lambda + w \tag{28}$$

Note that Eq. 27 is required since $\lambda$ is in constraint space and it should be in the positive direction that satisfies the constraint. Note also that Eq. 28 is required since constraint forces does not add energy to the system and thus should always be perpendicular to the direction of the constraint change.

## C   FROM SIMULATION TO REALITY: FUTURE WORK

The endoskeletal robots in this paper were confined to simulated tasks within a virtual world. Though significant effort was dedicated to ensuring that the simulated mechanics, simulated environments and simulated materials closely resembled physical reality and realizable morphologies (Table 7), it is in general difficult to guarantee that the behavior of a system optimized in simulation will transfer with sufficient fidelity to reality (Jakobi et al., 1995). It remains to determine the effect that universal controllers have on sim2real transfer, since sim2real of universal control has yet to be attempted. But because our approach simultaneously optimizes a population of many robots with differing topological, geometrical, sensory and motor layouts, it realizes the desired behavior in many unique ways and thereby provides many more opportunities for successful sim2real transfer compared to other methods that optimize a single design (Kriegman et al., 2020b). We also predict that the inherent capacity of a universal controller to generalize across these differing action and observation spaces will render the policy more transferable across certain differences between simulation and reality, compared to a bespoke single-morphology controller.

# D  OVERVIEW OF THE ENTIRE CO-DESIGN PIPELINE

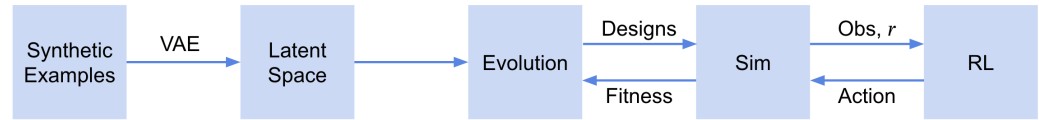

Figure 18: **Data flow.** Synthetic examples are used to encode the latent search space in which evolution operates. An evolving population of designs decoded from this space are simulated and used to train a controller. The population is updated based on the fitness of the designs.

# E  SYNTHETIC DATA GENERATION

---

**Algorithm 1** Procedural Generation of Synthetic Robot Body Plan

---

1: **Input:** Node range $[K_{\min}, K_{\max}]$, degrade ratio $\psi$
2: Randomly select number of nodes $k$ in $[K_{\min}, K_{\max}]$
3: **for** each node $i = 1$ to $k$ **do**
4:     Randomly initialize:
5:         Max children $C_i$
6:         Limb length $L_i$
7:         Rigid radius $r_{\text{rigid},i}$
8:         Soft radius $r_{\text{soft},i}$
9:     Create *StarNode i*
10: **end for**
11: Set node 1 as root (level 0)
12: **for** each node $i = 2$ to $k$ **do**
13:     Select parent node $p$ with lowest level that can attach more children
14:     Decrease limb length: $L_i = L_i \times \psi^{\text{level}_p}$
15:     Compute offset vector opposite to $p$'s existing children, blend with random unit vector
16:     From $p$, use $L_p$ and offset to find attachment point; use $L_i$ to place center of node $i$
17:     Attach node $i$ to parent node $p$
18: **end for**

---

Table 1: Hyperparameters and their ranges used in the procedural generation of synthetic robot body plans.

| Parameter | Range / Values | Description |
|---|---|---|
| *Global Parameters* | | |
| $K_{\min}$ | 3 | Minimum number of nodes (stars) per robot. |
| $K_{\max}$ | 8 | Maximum number of nodes (stars) per robot. |
| $\psi$ | 0.9 | Degradation ratio multiplier for limb length per hierarchical level. |
| *Node Parameters (for each node i)* | | |
| $C_i$ | $\{2, 3, 4\}$ | Maximum number of children (arms) for node $i$. |
| $L_i$ | $[0.1, 0.3]$ | Limb length for node $i$. |
| $r_{\text{rigid},i}$ | $[0.02, 0.05]$ | Radius of the rigid (bone) parts for node $i$. |
| $r_{\text{soft},i}$ | $[0.02, 0.07]$ | Radius of the soft (tissue) parts for node $i$. |

## F  MORE HYPERPARAMETERS

Table 2: VAE *encoder* architecture hyperparameters.

| Layer | Description |
|---|---|
| Input | Voxel input, $64^3 \times (k+2)$ |
| VRN0 | Voxception-ResNet (VRN), kernel size $3^3$, 32 channels |
| Block 1 | 5x VRN, 32 channels, downsample to $32^3$, 64 channels |
| Block 2 | 5x VRN, 64 channels, downsample to $16^3$, 128 channels |
| Block 3 | 5x VRN, 128 channels, downsample to $8^3$, 256 channels |
| Block 4 | 5x VRN, 256 channels, downsample to $4^3$, 512 channels |
| ResConv | Residual Conv, kernel size $3^3$, 512 channels |
| Max Pool | Max pooling, 512 channels |
| Linear | in: 512, out: 1024, GeLU |
| Linear-mu, Linear-logvar | mu: in: 512, out: 512, logvar: in: 512, out: 512 |
| Output | mu: 512, logvar: 512 |

Table 3: VAE *decoder* architecture hyperparameters.

| Layer | Description |
|---|---|
| Input | Embedding input, 512 |
| Linear | in: 512, out: 32768, reshaped to $4^3 \times 512$ |
| Block 1 | 5x VRN, 512 channels, ConvTranspose3D to $8^3$, 256 channels |
| Block 2 | 5x VRN, 256 channels, ConvTranspose3D to $16^3$, 128 channels |
| Block 3 | 5x VRN, 128 channels, ConvTranspose3D to $32^3$, 64 channels |
| Block 4 | 5x VRN, 64 channels, ConvTranspose3D to $64^3$, $k+2$ channels (voxel prediction) |
| Output | Voxel logits, $64^3 \times (k+2)$ |

Table 4: RL encoder architecture hyperparameters. The RL encoder is shared by the actor and critic, combining spatial pooling and graph transformer.

| Layer | Description |
|---|---|
| Input | $s^v$ (4 channels), $s^r$ (14 channels), $s^j$ (9 channels) |
| ResNet (voxels wise) | in: 4, out: 64 |
| Spatial Encoder-Decoder Net Peng et al. (2020) | Feature grid dim: $64^3 \times 64$ 
 UNet3D levels: 2 
 UNet3D groups: 8 
 UNet3D feature maps: 64 |
| TransformerConv 1 | node_in: 78 (64 + 14), node_out: 256, edge_in: 9 |
| TransformerConv 2 | node_in: 256, node_out: 512, edge_in: 9 |
| TransformerConv 3 | node_in: 512, node_out: 512, edge_in: 9 |
| Output | $\tilde{\mathbf{n}}_t$: 512 
 $\tilde{\mathbf{e}}_t$ : 1024 (by concatenate node features $\tilde{\mathbf{n}}_t$) |

Table 5: Actor architecture hyperparameters.

| Layer | Description |
|---|---|
| Input | $\tilde{\mathbf{e}}_t$ Outputs from RL encoder (see Table 4) |
| ResNet 1 | in: 1024, out: 256 |
| ResNet 2 | in: 256, out: 64 |
| ResNet 3 | in: 64, out: $\|\mathcal{A}\|$ |
| Output | $\mathbf{z}_t$: $\|\mathcal{A}\|$ |

Table 6: Critic architecture hyperparameters.

| Layer | Description |
|---|---|
| Input | $\tilde{\mathbf{n}}_t$ Outputs from RL encoder (see Table 4) |
| Max Pool | Max pooling, 512 channels |
| ResNet 1 | in: 512, out: 128 |
| ResNet 2 | in: 128, out: 32 |
| ResNet 3 | in: 32, out: 1 |
| Output | $V(s_t)$: 1 |

Table 7: Simulation configuration hyperparameters.

| Parameter | Value |
|---|---|
| *Environment* | |
| Rigid solver iterations | 10 |
| Baumgarte ratio ($\beta$) | 0.01 |
| Gravity acceleration | 9.81 m/s$^2$ |
| Control signal frequency | 10 Hz |
| Max joint motor torque | 6 N*m |
| Voxel size | 0.01 m |
| *Soft Material* | |
| Elastic modulus | $3 \times 10^4$ Pa |
| Density | 800 kg/m$^3$ |
| Poisson's ratio | 0.35 |
| Static friction | 1 |
| Dynamic friction | 0.8 |
| *Rigid Material* | |
| Density | 1500 kg/m$^3$ |
| Static friction | 1 |
| Dynamic friction | 0.8 |

Table 8: Reinforcement learning and evolution hyperparameters.

| Parameter | Value |
|---|---|
| *RL Hyperparameters* | |
| Actor learning rate | $6 \times 10^{-5}$ |
| Critic learning rate | $6 \times 10^{-5}$ |
| Discount ($\gamma$) | 0.9 |
| Entropy weight | 0.01 |
| GAE | Not used |
| Learning epochs | 20 |
| PPO train batch size | 64 |
| PPO train steps per epoch | 60 |
| $P_{\text{small}}$ | 0.1 [voxels] |
| $P_{\text{large}}$ | 10 [voxels] |
| Minimum movement threshold ($\delta$) | 0.001 m |
| *Population and Rollouts* | |
| Population size ($P_{\text{large}}$) | 64 |
| Clones per individual (rollout_num) | 2 |

Table 9: Symbols.

| Symbol | Meaning |
|---|---|
| *Simulation* | |
| $m$ | Mass |
| $x, y, z$ | Position of each axis in space |
| $\epsilon$ | Average strain of a soft voxel |
| $\mathbf{x}$ | Position in space |
| $\mathbf{v}$ | Linear velocity |
| $\boldsymbol{\omega}$ | Angular velocity of a rigid bone |
| $\mathbf{q}$ | Quaternion representing the orientation |
| $\boldsymbol{\theta}$ | Rotation in space |
| $\boldsymbol{\tau}$ | Moment or torque |
| $\boldsymbol{f}$ | Force |
| $X$ | Generalized position |
| $\dot{X}$ | Generalized velocity |
| $\ddot{X}$ | Generalized acceleration |
| $\dot{X}_{\text{old}}$ | Generalized velocity from the previous time step |
| $M$ | Generalized mass matrix |
| $K$ | Stiffness matrix of a beam |
| $I$ | Inertia |
| $F$ | Generalized force vector |
| $c(X)$ | Constraint function between two rigid bones |
| $J$ | Jacobian matrix |
| $\lambda$ | Lagrange multiplier |
| $A$ | Matrix in the LCP |
| $w$ | Term involving velocities and external forces in the LCP |
| $\Delta t$ | Time step size in simulation |
| $\beta$ | Constraint error correction rate |
| *Reinforcement Learning* | |
| $r$ | Reward |
| $s$ | State |
| $a$ | Action |
| $V$ | Value function |
| $\pi$ | Policy function |
| $\gamma$ | Discount factor |
| $\mathbf{z}$ | Action logits |
| $\mathcal{A}$ | Action space |
| $\mathbf{n}$ | Node features |
| $\mathbf{e}$ | Edge features |
| $\theta$ | Joint angle |
| $\mathbf{h}$ | Hinge axis direction of a joint |
| $\mathbf{d}_1, \mathbf{d}_2$ | Vectors from joint to the center of mass of connected bones |
| $\mathbf{u}$ | History movement vector |
| $\mathbf{v}$ | Movement vector between two time steps, or linear velocity |
| *Utilities* | |
| $\mathbb{I}$ | Indicator function |
| $E$ | Identity matrix |
| *Subscript and Superscript Conventions* | |
| $b$ | Beam (subscript) |
| ext | External (subscript) |
| $t$ | Time step in RL (subscript) |
| $v$ | Soft voxel (superscript) |
| $r$ | Rigid bone (superscript) |
| $j$ | Joint (subscript/superscript) |
| $l$ | Local pool (superscript) |

