# OpenReview forum: "Generating Freeform Endoskeletal Robots"
_ICLR.cc/2025/Conference — ICLR 2025 Spotlight_

### Official Review · Reviewer_wJ4t · 2024-10-29

**Soundness:** 4
**Presentation:** 3
**Contribution:** 4
**Rating:** 8
**Confidence:** 4

**Summary:**

This paper bridges the two opposing types of robots namely fully rigid and fully soft robots in current literature by introducing a novel multi-physics simulator of 3D endoskeletal robots that feature rigid, jointed bodies covered by soft materials. The authors employ a variational autoencoder trained on synthetic body plans to yield smooth and expressive latent representation of morphology. With the aid of a Graph Transformer-based modular control policy trained alongside morphological evolution, the authors showcase the feasibility of their simulation environment and latent representation to support the emergence of complex, lifelike morphologies capable of navigating various terrains. This work lays an important foundation for developing more sophisticated, biomimetic and free-form artificial organisms.

**Strengths:**

1. The paper introduces an open-source simulation environment of 3D endoskeletal robots with four predefined terrains. The voxel-based representation is conceptually simple and meanwhile offers great convenience for future work to benchmark their brain-body co-design algorithms. It also provides an open-endedness to design more complicated terrain maps for specific needs.

2. I appreciate that the authors meticulously designed a suite of control experiments to demonstrate the indispensability of morphological evolution. This is generally absent in existing brain-body co-design studies, which take morphology and control as a whole to evaluate task performance.

**Weaknesses:**

The introduction seems a bit too verbose. The authors are suggested to only keep the most essential information regarding current state of the art and their contributions, and relegate the remaining to Related Work or Appendix.

**Questions:**

1.Some previous studies have demonstrated that Graph Neural Networks tend to have difficulty with long-distance communication, as messages would be diluted in multiple hops [1]. Could you explain why you have chosen a Graph Transformer, instead of a conventional Transformer with full connection between modules (as in [1,2])?

[1] Kurin V, Igl M, Rocktäschel T, et al. My body is a cage: the role of morphology in graph-based incompatible control[C]//9th International Conference on Learning Representations. 2021.

[2] Gupta A, Fan L, Ganguli S, et al. Metamorph: learning universal controllers with transformers[C]//International Conference on Learning Representations. ICLR, 2022.

2.Is your simulation environment currently restricted to locomotion on various terrains? Does it or will it support manipulation tasks, such as carrying or pushing an object?

3.Could you explain why the population is cloned during policy training?

Thank you!

---

> ### Author Response · Authors · 2024-11-26
>
> We thank the reviewer for considering our work important and carefully reviewing our manuscript.
>
> >W1: The introduction seems a bit too verbose. The authors are suggested to only keep the most essential information regarding current state of the art and their contributions, and relegate the remaining to Related Work or Appendix.
>
> We share the reviewer’s concern about verbosity. But, because our paper synthesizes and builds on ideas within and across several fields of study (simulation, robotics, evolutionary design, reinforcement learning, etc.) there were many state-of-the-art methods that we felt inclined to discuss and cite up front in order to situate our work in the wider literature. Reviewer Le5n felt that “[t]he manuscript was a pleasure to read. The writing is clear and concise, with appropriate details provided in the extensive appendices.” So: with respect, we feel the introduction’s length is appropriate and indeed necessary to not only capture the essential background information regarding the state of the art and their contributions, but also the motivation for combining and moving beyond these works.
>
> >Q1: Some previous studies have demonstrated that Graph Neural Networks tend to have difficulty with long-distance communication, as messages would be diluted in multiple hops [1]. Could you explain why you have chosen a Graph Transformer, instead of a conventional Transformer with full connection between modules (as in [1,2])?
> >[1] Kurin V, Igl M, Rocktäschel T, et al. My body is a cage: the role of morphology in graph-based incompatible control[C]//9th International Conference on Learning Representations. 2021.
> >[2] Gupta A, Fan L, Ganguli S, et al. Metamorph: learning universal controllers with transformers[C]//International Conference on Learning Representations. ICLR, 2022.
>
> The graph transformer we used is analogous to the conventional transformers used in [1] and [2], but with graph connectivity injected as a bias into the attention matrix.
>
> Kurin et al. [1] use a global position scheme, but their learned attention mask is analogous to their respective graph connectivity. Gupta et al. [2] use a depth first traversing scheme to encode graph structure. This is why we cite this prior work as using a “graph transformer” for universal control in the introduction. In this work and in our own, the longest distance in the skeletal graph is less than 10 hops, and so we are unable to comment on the difficulty of long distance communication.
>
> The graph transformer we used was very easy to implement using an existing library (pytorch geometric) and did not require developing any custom functions. Our code is now available here: https://endoskeletal.github.io.
>
> >Q2: Is your simulation environment currently restricted to locomotion on various terrains? Does it or will it support manipulation tasks, such as carrying or pushing an object?
>
> We very much appreciate this question about the generality of our simulator. We have added an example of object manipulation (“dribbling a football” or soccer ball, if you prefer) in Fig. 8 in the appendix. Thank you for this suggestion! For convenience, we have copied and pasted the caption below:
>
> *Figure 8: Object manipulation. One of the designs that evolved for Upright Locomotion performs a new task: it pushes an object forward. This design was among several others that we sampled from the evolving population analyzed in Fig. 7, placed behind a soft sphere (teal), and actuated using the universal controller from the Optimized RL checkpoint in Fig. 7C. Videos of the resulting behaviors and the source code necessary to reproduce our results can be found on our project page: https://endoskeletal.github.io.*
>
> >Q3: Could you explain why the population is cloned during policy training?
>
> The rationale for including one clone of each design in the population was to reduce the likelihood of erroneously discarding a good design or preserving a bad design. However, we did not test if it would be better to include more than one clone or none at all. We have added an explanation for cloning to the paper. For convenience, we have copied and pasted the revision below:
>
> *We employed Proximal Policy Optimization (PPO; Schulman et al. (2017)) to train a single universal controller for an evolving population of 64 endoskeletal robots. A clone of each design in the population was created, yielding a batch of 128 designs. This last step is intended to broaden the evaluation of each design and so reduce the likelihood of erroneously discarding a good design.*

---

> > ### Comment · Reviewer_wJ4t · 2024-11-26
> > **Response to authors**
> >
> > Thank you for the reply.

---

### Official Review · Reviewer_uyFn · 2024-11-01

**Soundness:** 3
**Presentation:** 3
**Contribution:** 4
**Rating:** 8
**Confidence:** 3

**Summary:**

The authors designed and implemented a software library capable of generating three-dimensional freeform endoskeletal robots. Across various tasks, the researchers demonstrate the simulations’ ability to provide a means of benchmarking the evolutionary design and representational learning of complex embodies systems.

**Strengths:**

Provides an easy-to-use computational design of freeform endoskeletal robots and the library software for benchmarking the representations and evolution of these agents over time.

**Weaknesses:**

The paper's conclusion that "design evolution finds better designs but they do not imply that evolution (i.e. cumulative selection) is necessary" appears to contradict the study's methodology, which used reward values that created selection pressure across generations. Please clarify how this conclusion aligns with the reward-based selection used in the experiments, and if possible, provide evidence that random mutations alone could achieve similar results. We recommend rephrasing to better reflect the relationship between mutations and selection pressure while maintaining the valuable insights about design diversity.

**Questions:**

What insight may explain the evolution of the agents in these specific directions? Can we find similarities of this biologically?

---

> ### Author Response · Authors · 2024-11-26
>
> We thank the reviewer for taking the time to review our manuscript and helping us improve its clarity.
>
> >W1: The paper's conclusion that "design evolution finds better designs but they do not imply that evolution (i.e. cumulative selection) is necessary" appears to contradict the study's methodology, which used reward values that created selection pressure across generations. Please clarify how this conclusion aligns with the reward-based selection used in the experiments, and if possible, provide evidence that random mutations alone could achieve similar results. We recommend rephrasing to better reflect the relationship between mutations and selection pressure while maintaining the valuable insights about design diversity.
>
> Thank you for highlighting this confusing statement. Our claim about “the necessity of morphological evolution” is supported by the evolved design population outperforming the initial population, and evolution outperforming random search. We have rewritten this point to be more clear. For convenience, we have copied and pasted the new text below:
>
> *In both cases—starting policy training from scratch or from the RL checkpoint—the frozen evolved designs significantly outperformed the frozen initial designs (Fig. 7C). In both frozen design populations, the universal controller from the checkpoint immediately exhibits the best performance achieved after retraining from scratch. This shows that RL does not suffer catastrophic forgetting: the policy does not forget how to control the initial designs as the population evolves to contain different, better designs.*
>
> *These results suggest that morphological evolution yields better designs but they do not prove that evolution is necessary. It could be the case that good designs will simply arise by random mutations alone, without cumulative selection across discrete generations. As a final control experiment we tested this hypothesis by replacing evolutionary strategies with random search (purple curve in Fig. 7A,B). Random morphological search performed significantly worse than evolutionary strategies. This suggests that morphological evolution (mutation and selection) is indeed necessary.*

---

> > ### Author Response · Authors · 2024-11-26
> >
> > >Q1: What insight may explain the evolution of the agents in these specific directions? Can we find similarities of this biologically?
> >
> > We agree with the reviewer that this is the most interesting question, and it is one that we are currently investigating. In Results and briefly in Discussion we comment on the convergent evolution of snakes on flat ground and how legged locomotion arises with certain selection pressures, but we would be hesitant to draw any conclusions related to biological evolution. To better explain the evolution of the agents in these and other directions, we need a better understanding of the search space evolution operates in. To this end, we have conducted a new set of experiments in our revision (Figs. 13-15) that interrogate how morphological traits (phenotypes) are encoded by the latent genome, and how evolution can travel through genotype space to affect these traits in a modular fashion without disrupting other traits. For convenience, we have copied and pasted the new text relating to these experiments and captions of the new figures below:
> >
> > *[The latent genome] was found to contain tunable genes that encode specific morphological traits, such as body height, length, width, volume and bone count (Figs. 13-15).*
> >
> > *Figure 13: Screening for latent genes. We measured the Pearson correlation coefficient (ρ) of the 512 latent features and the body volume of one million randomly sampled designs (A). The set of latent features that are highly correlated with volume (|ρ| > 0.1) were extracted as “the gene for volume”, the effect of which can be increased or decreased on demand (B). The effect of tuning the gene for volume ±2σ is shown for the design decoded from the sample mean of the latent space (μ). The same procedure was repeated to identify and modulate genes for bone count (C, D), length (E, F), width (G, H) and height (I, J).*
> >
> > *Figure 14: Gene editing. The best design that evolved for Upright Locomotion was extracted (A) and the set of latent features found to be individually correlated with body volume—the gene for volume—was incrementally increased and the volume of the resulting design was measured (B-G). The entire latent genome is shown to the right of each decoded design, with features outside the volume gene grayed out. Positively correlated features (240, 390 and 471) were increased, and negatively correlated features (189 and 361) were decreased by 0.25σ in each row, yielding progressively thicker body parts within the evolved body shape.*
> >
> > *Figure 15: Programmable morphology. The best design that evolved for Upright Locomotion was extracted (from Fig. 6B,F) and the found genes for volume (A), bone count (B), length (C), width (D) and height (E) were individually modulated to demonstrate the precision with which evolved morphologies can be manually redesigned through “genetic engineering”. Some genes afford greater control over a specific morphological trait than others; however, this may be attributed to the simplicity of the procedure employed to screen for genes, which only captures pairwise linear associations between a manually-specified morphological trait and an individual latent feature.*

---

### Official Review · Reviewer_dYez · 2024-11-02

**Soundness:** 2
**Presentation:** 3
**Contribution:** 2
**Rating:** 6
**Confidence:** 5

**Summary:**

In this paper, the authors introduced a new brain-body design of freeform endoskeletal robots. The authors introduce a computational framework that integrates the generation of external and internal structures through a combination of 3D modeling, latent space encoding, reinforcement learning, and evolutionary strategies.

**Strengths:**

1. The authors integrated 3D modeling, latent space encoding, reinforcement learning, and evolutionary strategies into a cohesive framework for robot design.

2. Optimizing in the latent space significantly reduces the computational burden of the co-design process.

3. The use of a universal controller and simultaneous optimization of morphology and control through reinforcement learning demonstrates the robustness and adaptability of the approach.

4. The research topic of this paper is interesting and significant.

**Weaknesses:**

1. While the paper focus on the co-design problem of endoskeletal robots, which is innovative, the main contribution of this paper seems to be not clear. Besides, the analysis of scalability and the potential real-world applications are needed.

2. It would be better to have a picture to describe the whole co-design pipeline of the method.

3. The whole co-design pipeline (Learning the latent space description-> Training a universal controller via RL-> Optimizing morphology through EA) is not new. The most important innovation of the article seems to turn out to be the modeling of endoskeletal robots.

4. The authors utilize multiple advanced techniques, which may lead to high computational costs and complexity, might limit practical applications.

5. The latent space encoding, while effective, may lack interpretability, making it difficult to understand the underlying principles that guide the design evolution.

**Questions:**

1. Could the authors provide more insights into the latent space encoding? Specifically, how do specific regions of the latent space correspond to different types of robot designs?

2. Can the proposed method find endoskeletal robot morphologies which can generalize to several tasks?

3. How does the proposed method compare to other state-of-the-art approaches (Such as transform2act, ea+rl bi-level optimization) in terms of design diversity, performance, and computational efficiency?

4. What are the potential real-world applications of the robots designed by your methods? How feasible is the transition from simulated designs to real-world robotic systems? What are the main challenges in this transition?

**Details Of Ethics Concerns:**

Not available.

---

> ### Author Response · Authors · 2024-11-26
>
> >W1: While the paper focus on the co-design problem of endoskeletal robots, which is innovative, the main contribution of this paper seems to be not clear. Besides, the analysis of scalability and the potential real-world applications are needed.
>
> We thank the reviewer for considering our work to be innovative and taking the time to provide constructive feedback. We now provide more results and discussion related to the scalability of our approach (in our responses to Q2 and Q3, below) and potential for real world applications (in our response to Q4).
>
> >W2: It would be better to have a picture to describe the whole co-design pipeline of the method.
>
> We have added a simple diagram of the whole co-design design pipeline to the appendix of the paper (Fig. 18). We are brainstorming ways to include this directly into Fig. 1 in a way that is complementary to Fig. 2, which contains a high level summary of the pipeline. For now we include this new, minimal sketch of the pipeline’s dataflow in the appendix. For convenience, we have copied and pasted the caption of this figure below:
>
> *Figure 18: Synthetic examples are used to encode the latent search space in which evolution operates. An evolving population of designs decoded from this space are simulated and used to train a controller. The population is updated based on the fitness of the designs.*
>
> >W3: The whole co-design pipeline (Learning the latent space description-> Training a universal controller via RL-> Optimizing morphology through EA) is not new. The most important innovation of the article seems to turn out to be the modeling of endoskeletal robots.
>
> We agree that the most important innovation of our work is the simulation, which was built from the ground up to support freeform generation and control of morphological complex robots. However, we respectfully disagree with the reviewer regarding the novelty of our co-design pipeline. As detailed in the introduction, although each element in the co-design pipeline has been demonstrated individually in simple body plans, we show for the first time their synthesis, and we do so in complex body plans.
>
> >W4: The authors utilize multiple advanced techniques, which may lead to high computational costs and complexity, might limit practical applications.
>
> This is a valid concern, however, the highest computational cost comes from training the VAE, which we now provide for others to use at no cost in the github repo linked at the top of our project page (https://endoskeletal.github.io).

---

> > ### Author Response · Authors · 2024-11-26
> >
> > >W5: The latent space encoding, while effective, may lack interpretability, making it difficult to understand the underlying principles that guide the design evolution.
> >
> > >Q1: Could the authors provide more insights into the latent space encoding? Specifically, how do specific regions of the latent space correspond to different types of robot designs?
> >
> > We enjoyed attempting to answer this question. To do so, we conducted a new set of experiments (Figs. 13-15) that investigate the interpretability of the latent space encoding. For convenience, we have copied and pasted the new text relating to these experiments and captions of the new figures below:
> >
> > *[We] demonstrate the learned representation's structural and functional coherence, expressivity, smoothness (Fig. 5) and interpretability (Figs. 13-15).*
> >
> > *The learned latent space is smooth and expressive (Fig. 5) and was found to contain tunable genes that encode specific morphological traits, such as body height, length, width, volume and bone count (Figs. 13-15).*
> >
> > *Figure 13: Screening for latent genes. We measured the Pearson correlation coefficient (ρ) of the 512 latent features and the body volume of one million randomly sampled designs (A). The set of latent features that are highly correlated with volume (|ρ| > 0.1) were extracted as “the gene for volume”, the effect of which can be increased or decreased on demand (B). The effect of tuning the gene for volume ±2σ is shown for the design decoded from the sample mean of the latent space (μ). The same procedure was repeated to identify and modulate genes for bone count (C, D), length (E, F), width (G, H) and height (I, J).*
> >
> > *Figure 14: Gene editing. The best design that evolved for Upright Locomotion was extracted (A) and the set of latent features found to be individually correlated with body volume—the gene for volume—was incrementally increased and the volume of the resulting design was measured (B-G). The entire latent genome is shown to the right of each decoded design, with features outside the volume gene grayed out. Positively correlated features (240, 390 and 471) were increased, and negatively correlated features (189 and 361) were decreased by 0.25σ in each row, yielding progressively thicker body parts within the evolved body shape.*
> >
> > *Figure 15: Programmable morphology. The best design that evolved for Upright Locomotion was extracted (from Fig. 6B,F) and the found genes for volume (A), bone count (B), length (C), width (D) and height (E) were individually modulated to demonstrate the precision with which evolved morphologies can be manually redesigned through “genetic engineering”. Some genes afford greater control over a specific morphological trait than others; however, this may be attributed to the simplicity of the procedure employed to screen for genes, which only captures pairwise linear associations between a manually-specified morphological trait and an individual latent feature.*

---

> > > ### Author Response · Authors · 2024-11-26
> > >
> > > >Q2: Can the proposed method find endoskeletal robot morphologies which can generalize to several tasks?
> > >
> > > Although inter-task generalization is somewhat outside the scope of our paper, we appreciate your question and we have conducted a new experiment (Fig. 8) to begin to answer it. For convenience, we have copied and pasted the caption below:
> > >
> > > *Figure 8: Object manipulation. One of the designs that evolved for Upright Locomotion performs a new task: it pushes an object forward. This design was among several others that we sampled from the evolving population analyzed in Fig. 7, placed behind a soft sphere (teal), and actuated using the universal controller from the Optimized RL checkpoint in Fig. 7C. Videos of the resulting behaviors and the source code necessary to reproduce our results can be found on our project page: https://endoskeletal.github.io.*
> > >
> > > >Q3: How does the proposed method compare to other state-of-the-art approaches (Such as transform2act, ea+rl bi-level optimization) in terms of design diversity, performance, and computational efficiency?
> > >
> > > It is not clear how to adapt Transform2act to grow freeform endoskeletal body parts. So, we compared the proposed method to EA+RL bi-level optimization as suggested. We summarize the results of this additional experiment in Fig. 17. For convenience, we have copied and pasted its caption below:
> > >
> > > *Figure 17: Universal and individual controllers. An otherwise equivalent experiment was conducted in which the universal controller shared by all 64 designs in the population was replaced with a bespoke individual controller for each design. As in Fig. 6, the peak fitness achieved by each design was averaged across the population before plotting the cumulative max (A). This statistic, which we term “best so far”, captures the progress of the entire population. Training 64 independent policies for 30 epochs required 44.8 hours on 4 NVIDIA H100 SXM GPUs. During the same time frame on the same hardware the universal controller can be trained for 451 epochs and thereby achieve much higher fitness. Over the first 30 epochs of learning, the performance of the universal controller was not statistically different from that of the individual controllers (B), and since design evolution was only advanced a single generation, the designs under universal control were visually similar to those with independent controllers, with both populations closely resembling the random initial population (Fig. 11).*
> > >
> > > >Q4: What are the potential real-world applications of the robots designed by your methods? How feasible is the transition from simulated designs to real-world robotic systems? What are the main challenges in this transition?
> > >
> > > To address your question, we have added a section to the appendix (Appx. C) immediately following our detailed description of the simulator (Appx. B). In this new section we briefly discuss the challenges and opportunities of real-world endoskeletal machines. For convenience, we have copied and pasted the new section below:
> > >
> > > *C. FROM SIMULATION TO REALITY: FUTURE WORK
> > > The endoskeletal robots in this paper were confined to simulated tasks within a virtual world. Though significant effort was dedicated to ensuring that the simulated mechanics, simulated environments and simulated materials closely resembled physical reality and realizable morphologies (Table 7), it is in general difficult to guarantee that the behavior of a system optimized in simulation will transfer with sufficient fidelity to reality (Jakobi et al., 1995). It remains to determine the effect that universal controllers have on sim2real transfer, since sim2real of universal control has yet to be attempted. But because our approach simultaneously optimizes a population of many robots with differing topological, geometrical, sensory and motor layouts, it realizes the desired behavior in many unique ways and thereby provides many more opportunities for successful sim2real transfer compared to other methods that optimize a single design (Kriegman et al., 2020b). We also predict that the inherent capacity of a universal controller to generalize across these differing action and observation spaces will render the policy more transferable across certain differences between simulation and reality, compared to a bespoke single-morphology controller.*
> > >
> > > *References:*
> > >
> > > *Jakobi et al. “Noise and the reality gap: The use of simulation in evolutionary robotics.” In Proceedings of the European Conference on Artificial Life (ECAL), pp.704–720, 1995.*
> > >
> > > *Kriegman et al. “Scalable sim-to-real transfer of soft robot designs.” In Proceedings of the International Conference on Soft Robotics (RoboSoft), pp. 359–366, 2020b.*

---

> > > > ### Comment · Reviewer_dYez · 2024-11-30
> > > > **Thank you**
> > > >
> > > > Thank you for the feedback, I decide to raise my score. However, I still encourage the authors to consider more about how to use the proposed co-design pipeline to produce the physically achievable robots which can work in some real world scenarios.

---

> > > > > ### Author Response · Authors · 2024-11-30
> > > > >
> > > > > Thank you for pushing us to improve our paper, dYez. We are in fact now exploring how to use our pipeline to produce physical robots that work in the real world.
> > > > >
> > > > > And thank you for deciding to raise your score based on our improvements. However it appears that your original rating has not been updated.
> > > > >
> > > > > We are sorry to drag you back here :) but if you have a few minutes to log back in and update your rating we would appreciate it.

---

> > > > > > ### Comment · Reviewer_dYez · 2024-12-01
> > > > > >
> > > > > > DONE

---

### Official Review · Reviewer_Le5n · 2024-11-03

**Soundness:** 4
**Presentation:** 4
**Contribution:** 3
**Rating:** 8
**Confidence:** 4

**Summary:**

This work proposes (1) a simulation framework, (2) a generative model, and (3) a co-optimization procedure for endoskeletal robots that consist of both rigid bones and soft tissue, in contrast to rigid-only or soft-only bodies in many previous works. (1) The endoskeletal robots are simulated as a collection of elastic and rigid voxels in a 3D space, with type-specific constraint forces between them. (2) The learning-based generative model is trained by autoencoding a large dataset of synthetic training data (valid voxelized body plans) into a latent space. After training, the decoder generates voxelized body plans from this latent space, which are post-processed into a graph structure defining a simulated robot body. (3) The co-optimization procedure uses two nested training loops, in which the outer "evolutionary" loop searches through the latent space with CMA-ES to design better morphology, and the inner "lifetime" loop learns a reused actor/critic architecture with PPO to control an arbitrary morphology on control tasks. The agent is rewarded for forward locomotion across a variety of terrains.

**Strengths:**

**S1: Novelty.** Unlike many previous works that focus exclusively on rigid or soft bodies, this work combines them into a single framework. The manuscript provides an extensive related works section for situating this work within the field, which was helpful.

**S2: Clarity.** The manuscript was a pleasure to read. The writing is clear and concise, with appropriate details provided in the extensive appendices. The mathematical notation is simple and consistent. The figures are thoughtfully designed. Overall, great work!

**S3: Quality.** The methods are appropriately applied, and the experimental results are technically sound and informative (especially with the ablations).

Overall, I believe this is a solid work which I will recommend acceptance of, provided that my feedback below is adequately addressed.

**Weaknesses:**

**W1: Code.** I did not see a link to code, which is critical for reproducibility. Can the authors please provide this during the rebuttal?

**W2: Videos.** It will be useful to see qualitative results (i.e. videos) in addition to the quantitative reward plots. Currently, the reward values aren't contextualized, so it's had to tell if the behavior is actually good, only that it's improving.

**W3: Ablation of discrete action space.** It would be useful to show the comparison of discrete vs continuous action space (line 310), and provide some discussion. Is this a constraint from your simulation, or a learning issue? Wouldn't PPO also be able to control continuous action spaces, as it does in other related sim works?

**W4: Minor figure/text edits.**
- Figure 2 has nice aesthetics, but could be simplified. It has a lot of extraneous subpanel labels that aren't useful (Figure 5 too). Really the figure has 3 columns (initial, training, final), and the rows are instances of the same population. This message could be highlighted with in-figure annotations, like an arrow indicating the direction of training time.
- Figure 1 looks pretty, but what is its purpose? It looks like a bunch of molecules floating on a dark ocean, which was confusing as it primed me to think about fluid simulations, (but the sim only works on terrestrial envs, line 533). Also, the entire caption is one really loong sentence.
- Typos: line 376 sampled, line 398 the the, Fig 7 subpanel A label, Equation 10 star misplaced

**Questions:**

**Q1: Actor/critic inputs.** What's the rationale for conditioning the actor and critic on different state?

**Q2: Ablation of evolved designs.** On line 498, is that statement a mistake? It does not seem like Optimized RL is doing well with the initial designs. And I'm not sure how catastrophic forgetting is related, i.e. forgetting how to control a bad initial policy?

---

> ### Author Response · Authors · 2024-11-26
>
> We thank the reviewer for their close examination of our work and for this positive feedback.
>
> >W1: Code. I did not see a link to code, which is critical for reproducibility. Can the authors please provide this during the rebuttal?
>
> >W2: Videos. It will be useful to see qualitative results (i.e. videos) in addition to the quantitative reward plots. Currently, thereward values aren't contextualized, so it's had to tell if the behavior is actually good, only that it's improving.
>
> Yes. We should have included a link to code and videos in our original submission. Now we do: https://endoskeletal.github.io
>
> We thank the reviewer for their patience and hope they enjoy our videos and software.
>
> >W3: Ablation of discrete action space. It would be useful to show the comparison of discrete vs continuous action space(line 310), and provide some discussion. Is this a constraint from your simulation, or a learning issue? Wouldn't PPO also be able to control continuous action spaces, as it does in other related sim works?
>
> Yes, PPO is also able to control a continuous action space in our simulation. To demonstrate this we have conducted a new experiment (Fig. 16) comparing discrete vs continuous action space and provided some discussion. For convenience, we have copied and pasted the new text relating to these experiments and caption of the new figure below:
>
> *We found that a discrete action space greatly simplified and stabilized policy training compared to an otherwise equivalent continuous action space (Fig.16).*
>
> *Figure 16: Discrete and continuous action spaces. Peak (A) and average (B) fitness is plotted for the Upright Locomotion task, as in Fig. 7A and B. Two kinds of action spaces were compared: discrete (gray) and continuous (pink). The discrete action space is {-1.4 rad, -0.7 rad, 0 rad, 0.7 rad, 1.4 rad} and the continuous action space is [-1.4 rad, 1.4 rad], for all joints. In both panels, each line is a mean across the 64 designs in the population and the shaded regions are a 95% bootstrapped CI of the mean. Although both continuous and discrete action spaces are viable, a discrete set of actions greatly simplified and stabilized universal policy training under the tested conditions.*
>
> >W4: Minor figure/text edits. Figure 2 has nice aesthetics, but could be simplified. It has a lot of extraneous subpanel labels that aren't useful (Figure5 too). Really the figure has 3 columns (initial, training, final), and the rows are instances of the same population. This message could be highlighted with in-figure annotations, like an arrow indicating the direction of training time. Figure 1 looks pretty, but what is its purpose? It looks like a bunch of molecules floating on a dark ocean, which was confusing as it primed me to think about fluid simulations, (but the sim only works on terrestrial envs, line 533). Also,the entire caption is one really loong sentence.
> Typos: line 376 sampled, line 398 the the, Fig 7 subpanel A label, Equation 10 star misplaced
>
> We thank the reviewer for identifying ways to strengthen our paper. We have removed the pretty yet confusing graphics from Fig. 1 and corrected the typos you identified. We respectfully find the annotations A-Z in Figs. 2 and 5 to be helpful, but we are considering the alternatives you mentioned for the final, camera-ready figures.

---

> > ### Author Response · Authors · 2024-11-26
> >
> > >Q1: Actor/critic inputs. What's the rationale for conditioning the actor and critic on different state?
> >
> > Thank you for identifying this point of confusion. We have added a note in Sect. 2.3 clarifying that *“[t]he processed edge features are produced by concatenating processed node features across the two nodes connected by each edge”*. That is, the actor and critic share the same state (as encoded by the graph transformer) and differ only in their single-MLP output “heads”.
> >
> > >Q2: Ablation of evolved designs. On line 498, is that statement a mistake? It does not seem like Optimized RL is doing well with the initial designs. And I'm not sure how catastrophic forgetting is related, i.e. forgetting how to control a bad initial policy?
> >
> > Thank you for highlighting these confusing statements. We have sharpened these claims in the manuscript and copied and pasted the new text below for convenience:
> >
> > *In both cases—starting policy training from scratch or from the RL checkpoint—the frozen evolved designs significantly outperformed the frozen initial designs (Fig. 7C). In both frozen design populations, the universal controller from the checkpoint immediately exhibits the best performance achieved after retraining from scratch. This shows that RL does not suffer catastrophic forgetting: the policy does not forget how to control the initial designs as the population evolves to contain different, better designs.*
> >
> > *These results suggest that morphological evolution yields better designs but they do not prove that evolution is necessary. It could be the case that good designs will simply arise by random mutations alone, without cumulative selection across discrete generations. As a final control experiment we tested this hypothesis by replacing evolutionary strategies with random search (purple curve in Fig. 7A,B). Random morphological search performed significantly worse than evolutionary strategies. This suggests that morphological evolution (mutation and selection) is indeed necessary.*

---

### Meta-Review · Area_Chair_NWNT · 2024-12-23

**Metareview:**

This paper presents a novel framework for designing freeform endoskeletal robots by integrating morphological evolution and reinforcement learning. It is a good attempt bridging robotics, evolutionary biology, and machine learning. Its experiments showed that evolved designs outperform randomly generated ones. The authors conducted extensive experiments across various task environments, validating the effectiveness and practical applicability of their methods.

The AC read the paper, reviews, and discussions and considered it a well qualified paper for ICLR 2025.

**Additional Comments On Reviewer Discussion:**

After rebuttals/clarifications, the reviewers increased the evaluations more towards acceptance.

---

### Decision · Program_Chairs · 2025-01-22

Accept (Spotlight)